# Endothelial TGF-β signaling instructs smooth muscle cell development in the cardiac outflow tract

Giulia LM Boezio[1], Anabela Bensimon-Brito[1], Janett Piesker[2], Stefan Guenther[3], Christian SM Helker[1†*], Didier YR Stainier[1*]

[1]Department of Developmental Genetics, Max Planck Institute for Heart and Lung Research, Bad Nauheim, Germany; [2]Scientific Service Group Microscopy, Max Planck Institute for Heart and Lung Research, Bad Nauheim, Germany; [3]Bioinformatics and Deep Sequencing Platform, Max Planck Institute for Heart and Lung Research, Bad Nauheim, Germany

**Abstract** The development of the cardiac outflow tract (OFT), which connects the heart to the great arteries, relies on a complex crosstalk between endothelial (ECs) and smooth muscle (SMCs) cells. Defects in OFT development can lead to severe malformations, including aortic aneurysms, which are frequently associated with impaired TGF-β signaling. To better understand the role of TGF-β signaling in OFT formation, we generated zebrafish lacking the TGF-β receptor Alk5 and found a strikingly specific dilation of the OFT: *alk5-/-* OFTs exhibit increased EC numbers as well as extracellular matrix (ECM) and SMC disorganization. Surprisingly, endothelial-specific *alk5* overexpression in *alk5-/-* rescues the EC, ECM, and SMC defects. Transcriptomic analyses reveal downregulation of the ECM gene *fibulin-5,* which when overexpressed in ECs ameliorates OFT morphology and function. These findings reveal a new requirement for endothelial TGF-β signaling in OFT morphogenesis and suggest an important role for the endothelium in the etiology of aortic malformations.

*For correspondence:
christian.helker@biologie.uni-marburg.de (CSMH);
Didier.Stainier@mpi-bn.mpg.de (DYRS)

Present address: †Philipps University Marburg, Faculty of Biology, Cell Signaling and Dynamics, Marburg, Germany

## Introduction

The cardiovascular system is essential to deliver blood to the entire organism. Within the heart, however, the high pressure deriving from ventricular contractions needs to be buffered to avoid damage to the connecting vessels. The cardiac outflow tract (OFT), located at the arterial pole of the heart, fulfills this role and is a vital conduit between the heart and the vascular network (*Kelly and Buckingham, 2002*; *Sugishita et al., 2004*). Its importance is confirmed by the fact that errors in OFT morphogenesis lead to almost 30% of all congenital heart defects (CHD) in humans (*Neeb et al., 2013*). These malformations include defects in the alignment and septation of the OFT, such as persistent truncus arteriosus (PTA), transposition of the great arteries and overriding aorta, as well as coarctation or dilation of the arteries exiting the heart (*Neeb et al., 2013*; *Anderson et al., 2016*). However, the etiology of these malformations remains unclear, due to their multifactorial causes and the complex interplay between the different cell types in the developing heart.

In all vertebrates, the development of the OFT starts with the formation of a simple tube lined by endothelial cells (ECs) and surrounded by myocardium, both derived from late-differentiating second heart field (SHF) progenitors (*Mjaatvedt et al., 2001*; *Kelly and Buckingham, 2002*; *Meilhac et al., 2014*). Later, this tube becomes surrounded by smooth muscle cells (SMCs) of SHF and neural crest origins (*Anderson et al., 2003*; *Rothenberg et al., 2003*; *Waldo et al., 2005b*; *Anderson et al., 2016*). Eventually, in mammals, the OFT undergoes septation, cushion formation, and rotation, giving rise to the mature structure that forms the trunk of the great arteries (*Rothenberg et al., 2003*;

*Bajolle et al., 2006*). In zebrafish, the recruitment of SHF progenitors from the anterior lateral plate mesoderm to the OFT starts around 26 hours post-fertilization (hpf) and proceeds in multiple waves (*Hami et al., 2011*; *Zhou et al., 2011*; *Jahangiri et al., 2016*; *Paffett-Lugassy et al., 2017*; *Felker et al., 2018*). Starting at 54–56 hpf, a few immature SMCs expressing elastin appear at the arterial pole of the heart (*Grimes et al., 2006*; *Zhou et al., 2011*; *Jahangiri et al., 2016*; *Felker et al., 2018*). From 72 hpf onwards, most of the OFT is surrounded by SMCs expressing elastin, fibronectin and aggrecan (*Miao et al., 2007*; *Rambeau et al., 2017*; *Duchemin et al., 2019*), although they still lack the expression of specific markers. Eventually, a few neural crest cells also colonize the region (*Cavanaugh et al., 2015*). In zebrafish, the OFT does not septate and forms what is considered a 'subsidiary chamber' – the bulbus arteriosus (*Grimes et al., 2006*; *Grimes and Kirby, 2009*; *Grimes et al., 2010*; *Knight and Yelon, 2016*).

In terms of the cellular contributions to OFT development, most of the attention has been focused on the SMCs, cardiac neural crest, and CMs (*Kelly and Buckingham, 2002*; *Buckingham et al., 2005*; *Waldo et al., 2005a*; *Waldo et al., 2005b*). However, given the role of the endothelium in the formation of other structures including the cardiac ventricles (*Lee et al., 1995*; *Meyer and Birchmeier, 1995*; *Rasouli and Stainier, 2017*; *Rasouli et al., 2018*), ECs are also likely to play major roles in OFT formation.

Multiple signaling pathways including BMP, Notch, FGF, Wnt, and TGF-β have been implicated in OFT development (*Neeb et al., 2013*). In particular, mutations in several of the *TGF-β* family members have been associated with severe congenital heart malformations, including PTA and aneurysm of the great vessels (*Todorovic et al., 2007*; *Gillis et al., 2013*; *Takeda et al., 2018*). Despite the clear importance of TGF-β signaling in the development and homeostasis of the OFT and connecting vessels, the molecular mechanisms underlying these defects remain elusive, due to the context-dependent and controversial role of this pathway which exhibits complex interactions with other signaling pathways (*Massagué, 2012*; *Cunha et al., 2017*; *Goumans and Ten Dijke, 2018*; *Zhang, 2018*). In fact, loss-of-function mutations in many *TGF-β* genes lead to an unexpected hyperactivation of downstream signaling after the initials manifestation of the symptoms (*Jones et al., 2009*; *Lin and Yang, 2010*; *Mallat et al., 2017*; *Takeda et al., 2018*).

Activin receptor-like kinase 5 (Alk5, aka Tgfbr1) is the main type I receptor of the TGF-β signaling pathway. In mouse, *Alk5* expression is present specifically in the great arteries and the heart, enriched in the SMC layer of the aorta (*Seki et al., 2006*). A global *Alk5* mutation in mouse leads to embryonic lethality due to brain hemorrhage and presumed cardiac insufficiency (*Carvalho et al., 2007*). Notably, these phenotypes are recapitulated by the EC-specific deletion of *Alk5* (*Sridurongrit et al., 2008*), although their cause remains unclear. Conversely, despite its reported expression pattern, loss of *Alk5* in mouse cardiomyocytes, pericytes or SMCs does not lead to any obvious defects during the development of the heart or great vessels (*Sridurongrit et al., 2008*; *Dave et al., 2018*).

Despite the evidence for a potential role for ALK5 in the endothelium (*Sridurongrit et al., 2008*), most studies in OFT and aortic pathologies, such as aortic aneurysms, have been focused on the role of TGF-β signaling in SMCs and not ECs (*Choudhary et al., 2009*; *Guo and Chen, 2012*; *Gillis et al., 2013*; *Yang et al., 2016*; *Takeda et al., 2018*). In particular, aneurysms have been described as a weakening of the aortic wall due to SMC-specific defects, resulting in the dissection of the vessel (*Takeda et al., 2018*). Only recently have a few studies started considering the endothelium as a potential therapeutic target in aortic pathologies (*van de Pol et al., 2017*; *Sun et al., 2018*). The close proximity of ECs with SMCs makes their cross-talk essential for aortic development and homeostasis (*Lilly, 2014*; *Stratman et al., 2017*; *Segers et al., 2018*). However, the early lethality of the *Alk5* global and EC-specific KO mice prevents a deeper investigation of the role of this gene in heart and OFT development.

The use of zebrafish as a model system can help overcome the issues associated with early embryonic lethality resulting from severe cardiovascular defects (*Stainier and Fishman, 1994*). Moreover, this system allows a detailed *in vivo* analysis of the phenotype and the assessment of cardiovascular function during embryogenesis thanks to its amenability to live imaging. Overall, due to the conserved features of OFT development amongst vertebrates (*Grimes and Kirby, 2009*; *Grimes et al., 2010*), the zebrafish could help one to obtain new insights into the role of TGF-β signaling in OFT development and disease.

Here, we generated a zebrafish *alk5* mutant and observed a severe dilation of the developing OFT. We show that this phenotype results from early defects in EC proliferation, followed by aberrant SMC proliferation and organization. Live imaging and transcriptomic analyses further reveal an Alk5-dependent alteration in extracellular matrix (ECM) composition. Notably, we show that restoring Alk5 in the endothelium is sufficient to rescue the OFT phenotype, including the SMC organization defects. Moreover, we identify the ECM gene *fibulin-5* (*fbln5*) as a target of Alk5 signaling able to partially rescue the *alk5* mutant phenotype when overexpressed in ECs, providing a new therapeutic target for various aortic malformations.

## Results

### Loss of Alk5 causes specific defects in cardiac OFT formation

Mammalian *Tgfbrl/Alk5* has two paralogs in zebrafish, *alk5a* and *alk5b*, and from multiple datasets (*Pauli et al., 2012*; *Yang et al., 2013*; *Gauvrit et al., 2018*; *Mullapudi et al., 2018*), we found *alk5b* to be the highest expressed paralog during embryogenesis. To investigate its expression, we performed *in situ* hybridization and generated a transgenic reporter line, *TgBAC(alk5b:EGFP)*, by bacterial artificial chromosome (BAC) recombineering. We detected *alk5b* expression in the neural tube starting at 30 hpf and in the gut at 72 hpf (*Figure 1—figure supplement 1A–D'*). Notably, in the developing cardiovascular system, *alk5b* reporter expression appears to be restricted to the heart (*Figure 1A,B*), as we could not detect GFP expression in any other vascular beds (*Figure 1—figure supplement 1B,B'*). Within the heart, we detected *alk5b* expression in the OFT (*Figure 1A–B''*), where it is localized in both ECs and SMCs (*Figure 1B', B''*).

In order to investigate Alk5 function, we used CRISPR/Cas9 technology to generate mutants for *alk5a* and *alk5b*. We obtained a 4 bp and a 8 bp deletion in *alk5a* and *alk5b*, respectively, each leading to the predicted generation of truncated proteins, lacking the kinase domain (*Figure 1—figure supplement 1E*). Moreover, *alk5a* and *alk5b* mutant mRNA levels are decreased in the respective mutant fish while the other paralog does not appear to be upregulated, suggesting mutant mRNA degradation but a lack of transcriptional adaptation (*El-Brolosy et al., 2019*; *Figure 1—figure supplement 1F*). Single *alk5a* and *alk5b* mutant larvae do not exhibit any gross morphological defects, other than the lack of inflation of the swim bladder in *alk5b-/-* larvae (*Figure 1—figure supplement 1G–I*). Therefore, to achieve a complete blockade of Alk5 signaling, we generated *alk5a*, *alk5b* double mutants (*alk5a-/-;alk5b-/-*), hereafter referred to as *alk5* mutants. Loss of Alk5 function does not lead to early developmental defects until 72 hpf, when *alk5-/-* larvae start exhibiting pericardial edema, more evident at 96 hpf (*Figure 1—figure supplement 1J*), suggesting defective cardiac function. By analyzing heart morphology in live *Tg(kdrl:EGFP) alk5-/-* embryos, we observed a specific increase in OFT width by 54 hpf (*Figure 1C–H*). Live imaging of beating hearts showed that, by 78 hpf, the dilation of the *alk5* mutant OFT between systole and diastole is more than twice as large as in wild type (*Figure 1I–K*; *Figure 1—videos 1* and *2*). This abnormal expansion of the OFT is accompanied by its inability to pump blood into the connecting vessels, leading to retrograde flow into the ventricle (*Figure 1—video 3*).

In 78 hpf wild-type zebrafish, the OFT is connected to the aortic arches by a single vessel, the ventral aorta (VA) (*Figure 1—figure supplement 1K*). *alk5-/-* animals fail to form this vessel; instead, they display two independent vessels from the OFT to the left and right aortic arches (*Figure 1—figure supplement 1L*). Furthermore, a few *alk5* mutant OFTs exhibited clear ruptures in their endothelial layer at 96 hpf (*Figure 1L,L'*; *Figure 1—video 4*). To investigate this phenotype, we injected dextran into the circulation and 7 out of 11 *alk5-/-* larvae displayed dextran accumulation between the ECs and SMCs in the OFT and in the interstitial space amongst SMCs, a phenotype not observed in *alk5+/+* larvae (n = 10; *Figure 1M,N*). Notably, the cardiovascular defects in *alk5* mutant fish are restricted to the OFT and the VA, while all other vascular beds appear morphologically unaffected. The diameter of the dorsal aorta (DA) appears unaffected in *alk5* mutants compared with wild-type siblings at 56 and 96 hpf (*Figure 1—figure supplement 1M*), and the atrium and ventricle appear correctly shaped at 78 hpf (*Figure 1—figure supplement 1N,O*). However, after the onset of the OFT phenotype, *alk5-/-* larvae also exhibit retrograde blood flow through the atrioventricular canal (1/6 *alk5* +/+, 9/10 *alk5-/-*; data not shown), despite an unaffected heart rate (*Figure 1—figure*

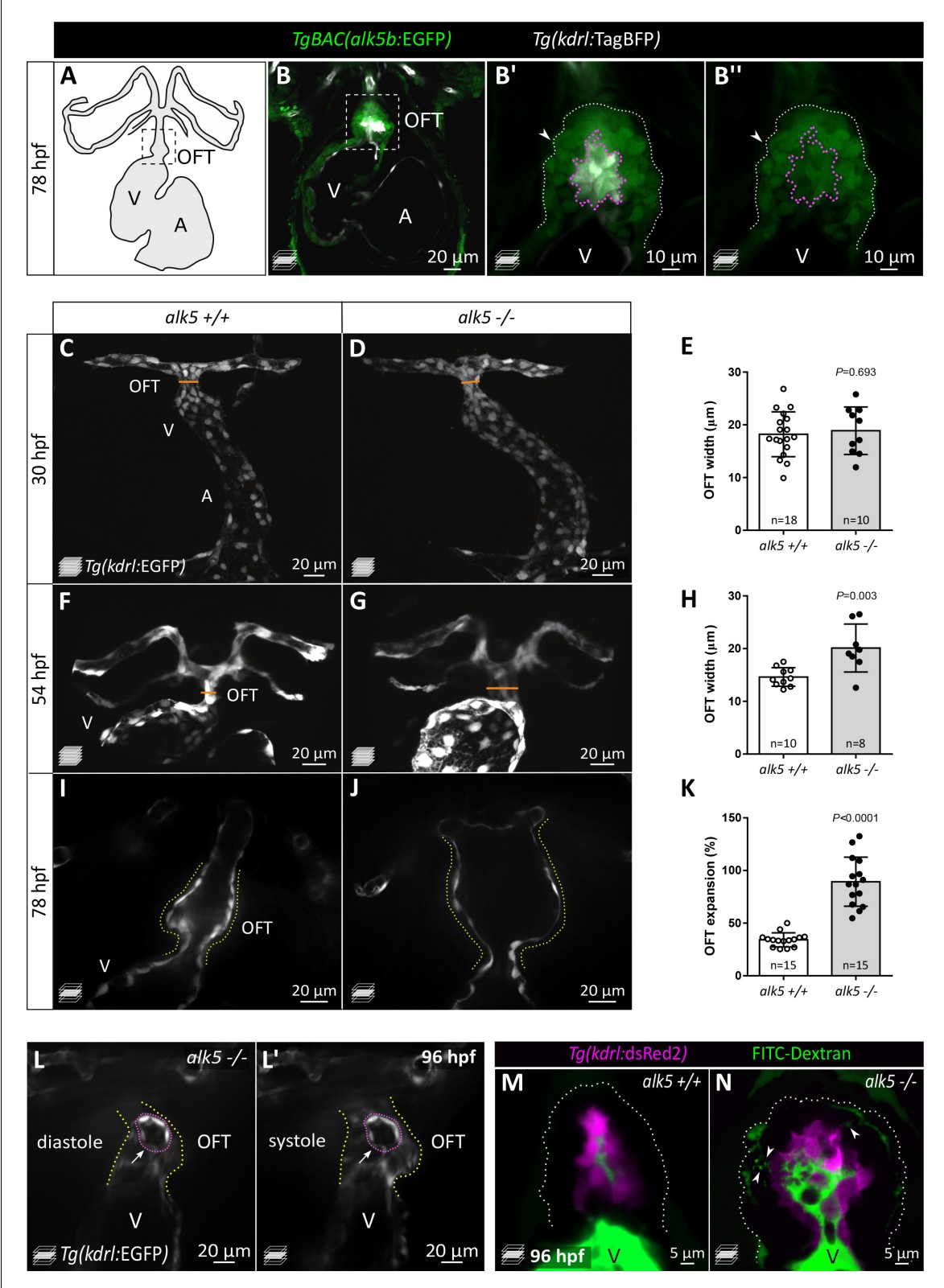

**Figure 1.** Loss of Alk5 function causes specific defects in cardiac OFT formation. (**A**) Schematic of the zebrafish heart and connecting vessels at 78 hpf; ventral view; black, endothelium/endocardium. (**B–B′′**) Confocal images showing *TgBAC(alk5b*:EGFP) expression (green) in the 78 hpf zebrafish heart. White, ECs; arrowheads, SMCs; boxed area shown in **B′** and **B′′**; white dotted lines outline the OFT; magenta dotted lines outline ECs. (**C–H**) Confocal images (**C, D, F, G**) and quantification (**E, H**) of OFT width in *alk5+/+* and *alk5-/-* embryos at 30 (**C–E**) and 54 (**F–H**) hpf. Orange line shows OFT width

*Figure 1 continued on next page*

*Figure 1 continued*

graphed in E and H. (I–K) Frames of confocal movies of beating hearts (I, J) and quantification of OFT expansion (K) at 78 hpf. For details about the quantifications, see Materials and methods 'Defining the landmarks of the OFT'. (L, L') Frames of confocal movies of 96 hpf *alk5-/-* beating hearts during ventricular diastole (L) and systole (L'). Pink dotted lines outline ECs surrounding the rupture; arrows point to the site of EC rupture; yellow dotted lines outline the OFT. (M, N) Confocal images of 96 hpf *alk5+/+* and *alk5-/-* OFTs showing the accumulation (arrowheads) of FITC-Dextran (green) between the SMCs in *alk5-/-* OFTs (7/11; *alk5+/+* 0/10). Magenta, ECs; dotted lines outline the OFT. (C–G) Maximum intensity projections. (B, I–N) Single confocal planes. (E, H, K) Plot values represent mean ± SD; p values from *t*-tests. A- atrium, V- ventricle. Scale bars: (B, C–L') 20 µm; (B', B'') 10 µm; (M, N) 5 µm. See also *Figure 1—figure supplement 1*.

The online version of this article includes the following video and figure supplement(s) for figure 1:

**Figure supplement 1.** *alk5* expression and function in zebrafish embryos and larvae.

**Figure 1—video 1.** Confocal video of the beating heart of a *Tg(kdrl:EGFP) alk5+/+* larva at 78 hpf, showing the OFT expansion.

https://elifesciences.org/articles/57603#fig1video1

**Figure 1—video 2.** Confocal video of the beating heart of a *Tg(kdrl:EGFP) alk5-/-* larva at 78 hpf, showing the OFT expansion.

https://elifesciences.org/articles/57603#fig1video2

**Figure 1—video 3.** Brightfield video of the beating heart of the *alk5+/+* and *alk5-/-* larvae shown in *Figure 1—videos 1* and *2* at 78 hpf, showing blood flow in the OFT.

https://elifesciences.org/articles/57603#fig1video3

**Figure 1—video 4.** Confocal video of the beating heart of a *Tg(kdrl:EGFP) alk5-/-* larva at 96 hpf, showing a hole in the endothelial lining of the OFT.

https://elifesciences.org/articles/57603#fig1video4

---

*supplement 1P*). Altogether, these defects lead to the lack of blood flow in the trunk by 120 hpf, and consequently, the death of *alk5-/-* larvae by 7 dpf.

Taken together, these results identify a previously unknown and specific requirement for Alk5 in OFT morphogenesis, structural integrity, and function.

## The Smad3 signaling axis is activated early in OFT ECs and downregulated in *alk5* mutants

Since Alk5 is the main type I receptor of the TGF-β signaling pathway, we wanted to investigate the pathway activity. Therefore, we analyzed the activation of Smad3 which, together with Smad2, is the main effector of TGF-β signaling upon its phosphorylation by the receptor. To this aim, we first took advantage of a Smad3-responsive element reporter line (*Tg(12xSBE:EGFP) Casari et al., 2014*) and observed an enriched GFP signal in the OFT at 24 and 75 hpf (*Figure 2—figure supplement 1A–B'*). In addition, we performed immunostaining on wild-type embryos and larvae for the phosphorylated form of Smad3 (p-Smad3). P-Smad3 immunostaining was detectable in the heart as early as 24 hpf (*Figure 2A,A'*), and it persisted until 75 hpf (*Figure 2—figure supplement 1C,C'*), exhibiting a pattern in the OFT consistent with the expression of the *Tg(SBE:EGFP)* reporter (*Figure 2—figure supplement 1D,D'*). Of note, at 24 hpf, the p-Smad3 OFT signal was strongly enriched in ECs compared with the surrounding cells (2 ± 0.5 folds higher in ECs), while this enrichment was lost by 75 hpf (*Figure 2B*). Moreover, we detected a 1.7 ± 0.3 fold higher p-Smad3 signal in OFT ECs compared with ECs in the rest of the heart (*Figure 2C*, *Figure 2—figure supplement 1E*). These data suggest that while *alk5b* reporter expression is broadly detected in the heart and in OFT ECs and SMCs (*Figure 1*), Alk5 signaling is more prominent in OFT ECs at early stages.

Next, we analyzed the effect of Alk5 loss on Smad3 signaling by performing p-Smad3 immunostaining on *alk5+/+* and *alk5-/-* embryos. As early as 24 hpf, that is, before the detection of a morphological defect, we observed a significant decrease in p-Smad3 immunostaining in *alk5-/-* OFTs in both ECs and surrounding cells (*Figure 2D–F*). To assess the requirement for Smad3 function downstream of Alk5 in the OFT, we treated *alk5+/+* and *alk5-/-* embryos with the Smad3 inhibitor SIS3 (*Jinnin et al., 2006*; *Dogra et al., 2017*) starting at 36 hpf and analyzed OFT morphology and function at 75 hpf (*Figure 2G–K*). Notably, Smad3 inhibition was sufficient to induce a mild OFT phenotype in *alk5* heterozygotes, also preventing the formation of the VA (*Figure 2G,H*), and to exacerbate the OFT defect of *alk5-/-* larvae (*Figure 2I–K*).

In addition, since Alk5 has been reported to stand at the cross-road between the TGF-β and BMP signaling pathways (*Grönroos et al., 2012*; *Holtzhausen et al., 2014*; *Aspalter et al., 2015*), we also addressed the effect of Alk5 loss on BMP signaling by immunostaining embryos for p-Smad1/5/8 (the BMP-specific Smads). However, we did not observe preferential activation of Smad1/5/8 in

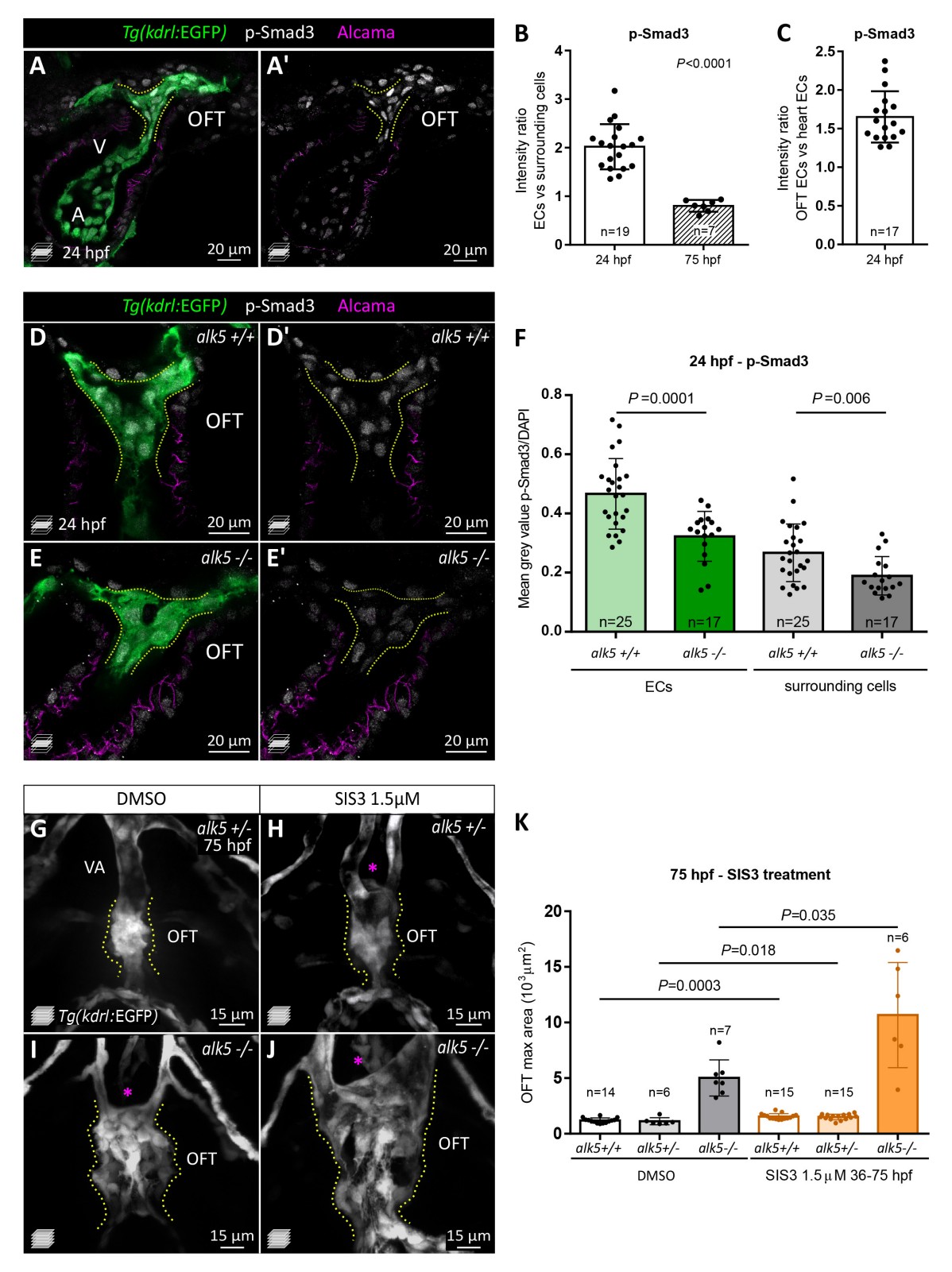

**Figure 2.** p-Smad3 is observed in OFT ECs at 24 hpf and reduced in *alk5* mutants. (**A, A'**) 24 hpf *Tg(kdrl:EGFP) alk5+/+* hearts immunostained for p-Smad3 (white). Green, ECs; magenta, CMs. (**B, C**) Intensity ratio of p-Smad3 immunostaining (normalized to DAPI) between ECs and surrounding cells at 24 and 75 hpf (**B**) and between OFT ECs and heart ECs in 24 hpf *alk5+/+* embryos (**C**). Every dot represents the ratio for one embryo. (**D–E'**) 24 hpf *Tg(kdrl:EGFP) alk5+/+* (**D**) and *alk5-/-* (**E**) hearts immunostained for p-Smad3 (white). (**F**) Quantification of p-Smad3 immunostaining (normalized to

*Figure 2 continued on next page*

*Figure 2 continued*

DAPI) in ECs and surrounding cells comparing 24 hpf *alk5+/+* and *alk5-/-* OFTs. (**G–J**) Confocal images of the OFT in 75 hpf *Tg(kdrl:EGFP) alk5+/-* (*alk5a-/-;alk5b+/-;* **G, H**) and *alk5-/-* (**I, J**) larvae treated with DMSO (**G, I**) or SIS3 (**H, J**) from 36 until 75 hpf. Asterisks point to the absence of the VA. (**K**) OFT maximum area in 75 hpf larvae treated with DMSO or SIS3. (**A–J**) Dotted lines outline the OFT ECs. (**B, C, F, K**) Plot values represent means ± SD; p values from Mann Whitney (**B, K**) and *t*-test (**F**). In K, p values refer to the comparisons highlighted. Scale bars: (**A, A', D–E'**) 20 µm; (**G–J**) 15 µm. See also *Figure 2—figure supplement 1* and *Figure 2—source data 1* and *2*.

The online version of this article includes the following source data and figure supplement(s) for figure 2:

**Source data 1.** Quantification of p-Smads immunostaining in 24 and 75 hpf wild-type animals.
**Source data 2.** Quantification of p-Smads immunostaining in 24 hpf *alk5+/+* and *alk5-/-* embryos.
**Figure supplement 1.** Evidence for p-Smad3 presence and function in OFT ECs at 24 hpf, but no evidence for p-Smad1/5/8 presence.

the wild-type OFT endothelium (*Figure 2—figure supplement 1F*) or any significant differences between *alk5+/+* and *alk5-/-* embryos at 24 hpf (*Figure 2—figure supplement 1G–I*).

Collectively, these data indicate that Alk5 promotes Smad3 activation in the OFT endothelium, while it does not seem to affect the BMP signaling axis.

## Alk5 restricts EC proliferation in the cardiac OFT and promotes EC displacement toward the ventral aorta

Given the increased size of the mutant OFT, we first asked whether this phenotype was accompanied by an increase in cell number. In *alk5+/+* animals, the average number of OFT ECs increases from 21 cells at 36 hpf to 45 cells at 72 hpf (*Figure 3A,B*). Consistent with the absence of a morphological phenotype, *alk5-/-* OFTs at 36 hpf were composed of an average of 22 ECs, similar to wild type (*Figure 3B*). However, the number of ECs in the mutant OFT diverged substantially over time, and by 72 hpf twice as many ECs were observed in *alk5-/-* compared with wild type (85 ECs; *Figure 3B*). To investigate the underlying cause of this increase in cell number, we performed EdU labeling to assess EC proliferation. In the *alk5-/-* OFTs, we found that, by 36 hpf, ECs were already more likely to undergo cell cycle reentry than ECs in *alk5+/+* OFTs (*Figure 3C–E*). This abnormal increase in the number of EdU$^+$ ECs in *alk5-/-* OFTs became even more pronounced at later stages (48–72 hpf; *Figure 3F–H*).

Together with an increased OFT size, *alk5* mutants also fail to form a VA. Therefore, we set out to investigate the behavior of ECs in the OFT and VA regions in embryos and larvae lacking Alk5 activity. We photoconverted OFT ECs in control and Alk5 inhibitor-treated larvae; treatment starting at 36 hpf with 2.5 µM of an Alk5-specific inhibitor caused increased OFT width at 54 and 78 hpf (*Figure 3—figure supplement 1A–F*) as well as VA patterning defects (*Figure 3—figure supplement 1D,E*), thus phenocopying *alk5* mutants. Using this Alk5 inhibitor with the *Tg(fli1a:Kaede)* line, we aimed to track the photoconverted ECs in the region of the OFT. Since the population of ECs which gives rise to the VA was not previously characterized, we photoconverted ECs at 54 hpf in different regions of the OFT: the proximal region close to the bulbo-ventricular (BV) valve (*Figure 3—figure supplement 1G,H*), and the most distal region between the aortic arches (*Figure 3—figure supplement 1J,K*). In wild-type fish, proximal ECs remained close to the photoconversion site up to 74 hpf (*Figure 3—figure supplement 1H,I*), whereas distal ECs were invariably found in the VA (*Figure 3I–J'*). In contrast, upon Alk5 inhibition, distal EC cells remained in the OFT (*Figure 3—figure supplement 1L*, *Figure 3K,K'*). By 74 hpf, these distal ECs in Alk5 inhibitor treated larvae were found closer to the original photoconversion site compared with control larvae (*Figure 3L*). To observe EC behavior at high resolution, we recorded time-lapse movies from 56 to 74 hpf following photoconversion (*Figure 3—videos 1* and *2*). While photoconverted ECs in control larvae extended rostrally and assumed an elongated shape (*Figure 3—video 1*), suggesting their movement toward the VA, ECs in Alk5 inhibitor-treated larvae did not appear to move from their original position (*Figure 3—video 2*).

Altogether, these data indicate that Alk5 plays a role in restricting EC proliferation in the OFT; it also promotes the formation of the ventral artery through a yet to be defined mechanism.

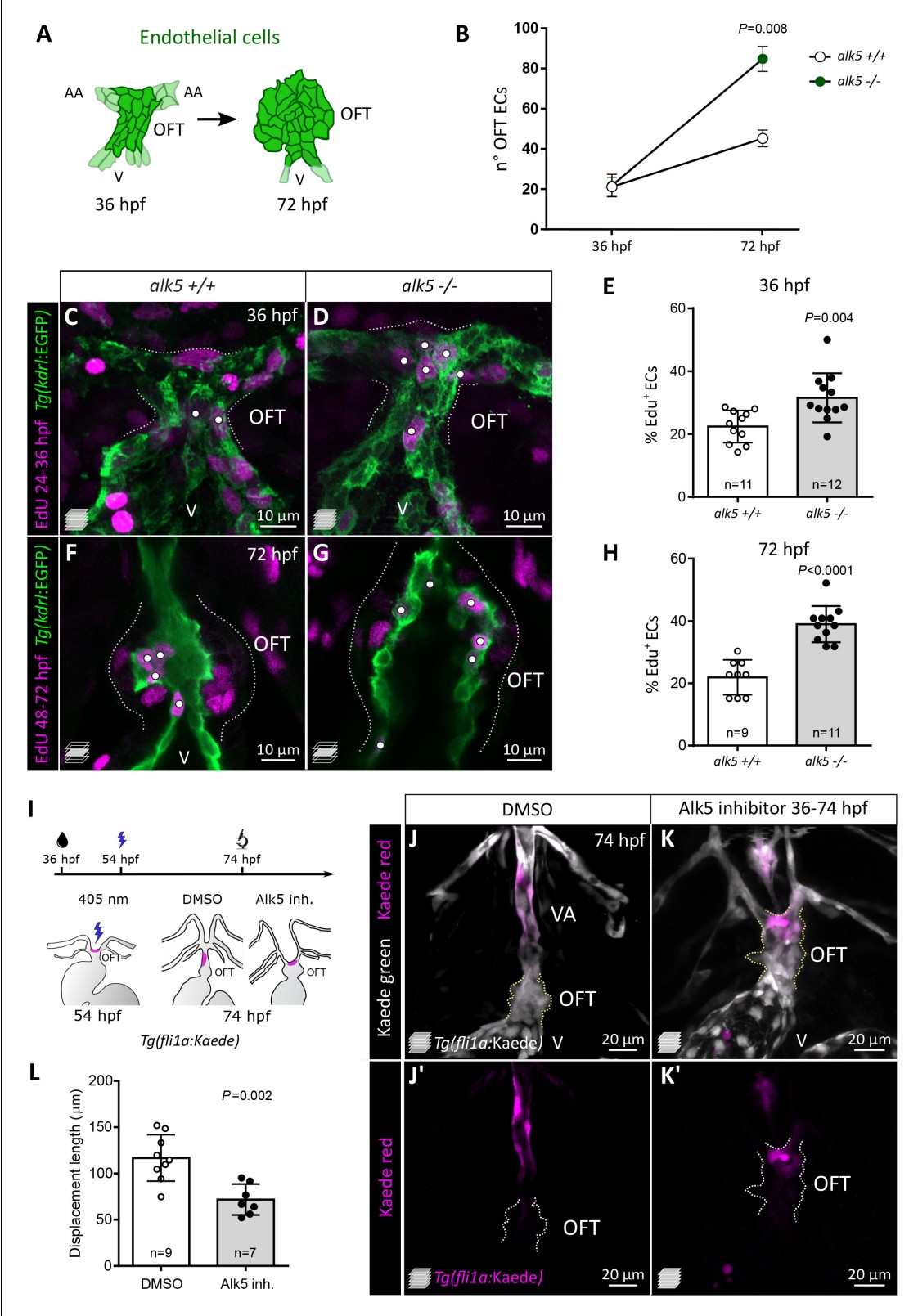

**Figure 3.** Alk5 restricts EC proliferation in the cardiac OFT and is required for ventral aorta formation. (**A**) Schematics of OFT ECs at 36 and 72 hpf. (**B**) Quantification of EC number (darker cells shown in A) in 36 and 72 hpf *alk5+/+* and *alk5-/-* OFTs (36 hpf: n = 11; 72 hpf: n = 6). (**C–H**) Confocal images (**C, D, F, G**) and quantification (**E, H**) of the percentage of EdU⁺ ECs in *Tg(kdrl:EGFP) alk5+/+* and *alk5-/-* OFTs. Dotted lines outline the OFT; white dots mark EdU⁺ ECs. All the other EdU⁺ cells in the OFT region are *kdrl*:EGFP⁻. (**I**) Protocol used for photoconversion experiment. (**J–K'**) Confocal

*Figure 3 continued on next page*

*Figure 3 continued*

images of the OFT in 74 hpf *Tg(fli1a:Kaede)* larvae treated with DMSO or Alk5 inhibitor. Magenta, photoconverted ECs; dotted lines outline the OFT. (L) Quantification of the distance covered by photoconverted ECs between 54 and 72 hpf in DMSO and Alk5 inhibitor-treated larvae. (C, D, J–K') Maximum intensity projections. (F, G) Single confocal planes. (B, E, H, L) Plot values represent means ± SD; p values from *t*-tests (E, H) or Mann Whitney (B, L). Scale bars: (C-G) 10 μm; (J–K') 20 μm. See also *Figure 3—figure supplement 1*.

The online version of this article includes the following video and figure supplement(s) for figure 3:

**Figure supplement 1.** Phenocopy of *alk5* mutants by Alk5 inhibitor treatment.

**Figure 3—video 1.** Time lapse-imaging of a *Tg(fli1a:Kaede)* fish treated with DMSO starting at 36 hpf and imaged from 56 to 74 hpf.

https://elifesciences.org/articles/57603#fig3video1

**Figure 3—video 2.** Time lapse-imaging of a *Tg(fli1a:Kaede)* fish treated with Alk5 inhibitor starting at 36 hpf and imaged from 56 to 74 hpf.

https://elifesciences.org/articles/57603#fig3video2

## Alk5 promotes the formation and stability of the OFT wall by regulating SMC proliferation and organization

During early larval stages, the OFT becomes covered by SMCs, which allow it to buffer the high blood pressure caused by ventricular contractions. In order to visualize SMCs, we used a pan-mural cell reporter line, *Tg(pdgfrb:EGFP)*, which labels pericytes and SMCs before they mature and start expressing established markers such as Acta2 (smooth muscle actin). We observed that the OFT endothelium in 75 hpf *alk5+/+* larvae is surrounded by an average of $90 \pm 2$ $pdgfrb^+$ cells, organized in two to three compact layers with limited extracellular space (*Figure 4A,A', C*). In contrast, *alk5-/-* larvae display a reduced number of $pdgfrb^+$ SMCs ($65 \pm 2, -27\%$), wider extracellular space, and disorganized cell layers around the OFT (*Figure 4B–C*). The reduction in the total number of SMCs in *alk5-/-* OFTs is likely caused by a proliferation defect, as confirmed by EdU incorporation experiments performed at early larval stages ($-53\%$; 48–72 hpf, *Figure 4D*). Furthermore, using TUNEL staining, we saw no evidence of SMC death by apoptosis (data not shown).

In order to form a compact yet elastic wall, the SMCs are embedded within a specialized extracellular matrix (ECM), which provides essential biomechanical support as well as signaling cues to the SMCs (*Raines, 2000*). To assess whether and how ECM structure and composition were affected in *alk5* mutants, we analyzed the localization of Elastin2 (Eln2; *Figure 4E–G*), a major ECM component in the OFT (*Miao et al., 2007*). We observed that in *alk5+/+* larvae 77.4% of SMCs were surrounded by a continuous layer of Eln2, while in *alk5-/-* larvae only 35.8% were (*Figure 4G*). Moreover, in *alk5-/-* OFTs Eln2 localizes in small disrupted clusters, rather than in a bundle of elastic fibers as observed in *alk5+/+* (*Figure 4E–F''*).

Next, we used transmission electron microscopy (TEM) to investigate OFT ultrastructure (*Figure 4H–K*). Even in a low-magnification view of *alk5-/-* OFTs, we could observe a greatly widened extracellular space surrounding the SMCs (*Figure 4I*). At higher magnification, we observed that the ECM in wild-type larvae consisted mainly of thin layers (*Figure 4J*), while it consisted of broader electron-negative spaces in *alk5* mutants (*Figure 4K*). Moreover, SMCs in the external layer of the mutant OFTs exhibited cytoplasmic inclusions (*Figure 4I,K*), such as electron-dense lysosomes (as confirmed by LysoTracker staining; *Figure 4L–M'*), and double-membraned vacuoles, resembling autophagosomes.

Overall, the loss of Alk5 function results in the formation of a defective SMC wall, including a structurally impaired ECM.

## Endothelial *alk5* overexpression is sufficient to restore cardiac OFT wall formation and function in *alk5* mutants

The earliest phenotype in *alk5* mutants is observed during the EC proliferation stages, therefore preceding the formation of the SMC wall. Thus, to investigate Alk5 requirement in the endothelium, we generated a transgenic line overexpressing *alk5b* specifically in ECs. We used the *fli1a* promoter to drive *alk5b* expression (*Figure 5A*) and validated the specificity of *Tg(fli1a:alk5b-mScarlet)* expression in endothelial and endocardial cells (*Figure 5—figure supplement 1A–A''*). We found that fish overexpressing *alk5b* in wild-type ECs reached adulthood, were fertile, and did not display obvious cardiovascular phenotypes (*Figure 5—figure supplement 1A*). Notably, when *alk5b* was overexpressed specifically in ECs in *alk5* mutants, it was sufficient to restore cardiac function, as indicated

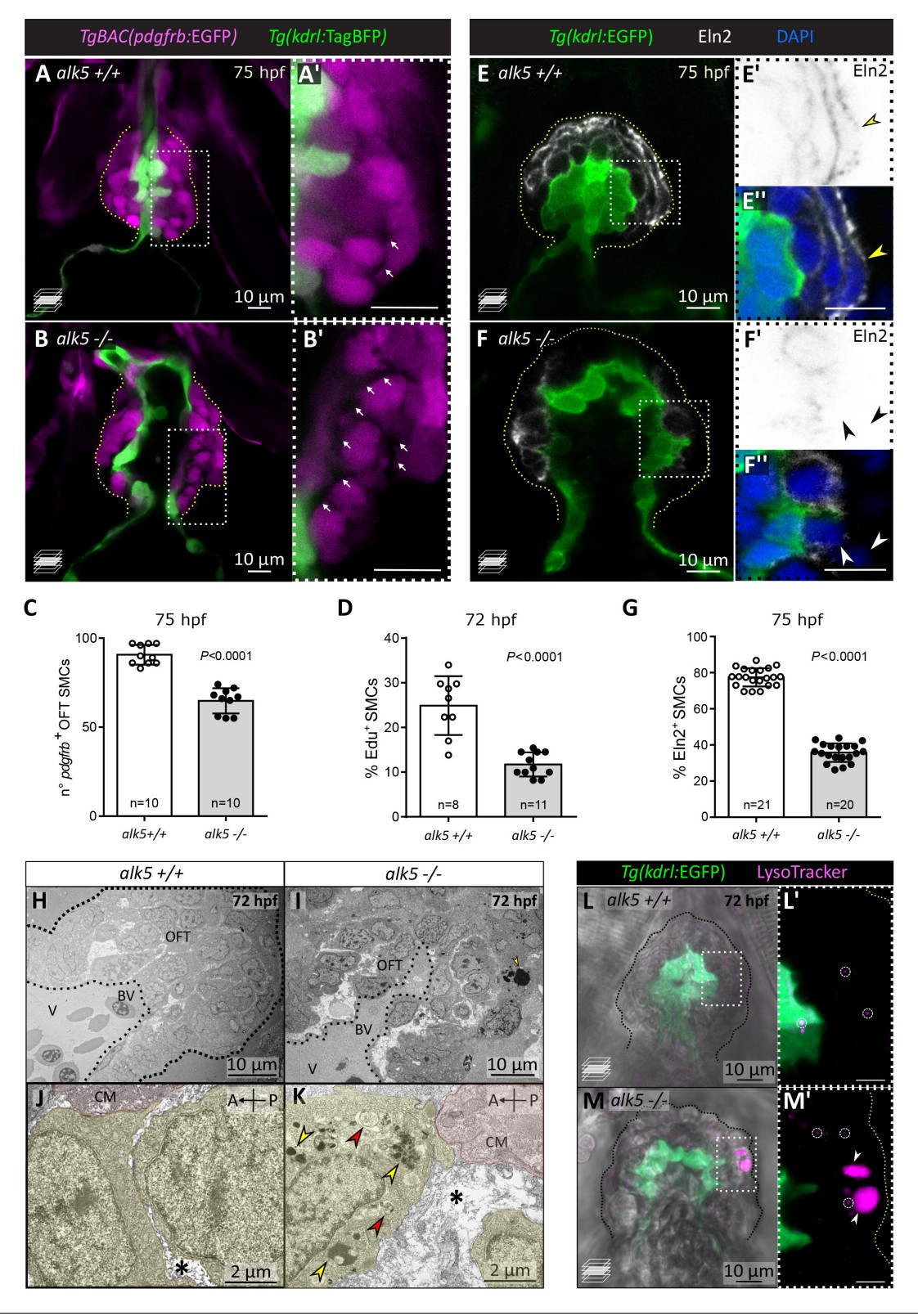

**Figure 4.** Alk5 regulates SMC and ECM organization in the cardiac OFT. (A–C) Confocal images (A–B') and quantification (C) of SMCs in 75 hpf *alk5+/+* and *alk5-/-* larvae. Magenta, SMCs; white arrows, extracellular space between SMCs; boxed areas are shown in A' and B'; dotted lines outline the OFT. (D) Percentage of EdU+ SMCs in 72 hpf *alk5+/+* and *alk5-/-* larvae. (E–F'') Confocal images of 75 hpf *alk5+/+* and *alk5-/-* larvae immunostained for Elastin2 (Eln2). White arrowheads, SMCs devoid of Eln2; yellow arrowheads, SMCs surrounded by Eln2 immunostaining; boxed areas shown in E', E'', F'

*Figure 4 continued on next page*

Figure 4 continued

and **F''**; images in **E'** and **F'** are shown with inverted colors; dotted lines outline the OFT. (G) Quantification of the percentage of SMCs surrounded by Eln2 immunostaining (per sagittal plane) at 75 hpf. (**H–K**) TEM images of 72 hpf *alk5+/+* and *alk5-/-* mutant OFTs at different magnifications (n = 3 for each genotype). Yellow, SMCs; red, cardiomyocytes close to the BV canal; asterisks, extracellular space; arrows, electron-dense (yellow) and double-membraned (red) vacuoles; dotted lines outline the OFT. (**L–M'**) Confocal images of *alk5+/+* (n = 5) and *alk5-/-* (n = 9) animals treated with LysoTracker, labeling lysosomes (small, circles; big, arrowheads); boxed areas are shown in **L'** and **M'**; dotted lines outline the OFT. (**C, D, G**) Plot values represent means ± SD; p values from *t*-tests. A- anterior, P- posterior, BV- bulbo-ventricular canal, CM- cardiomyocyte. Scale bars: , (**A-I, L–M'**) 10 µm; (**J, K**) 2 µm.

by the lack of pericardial edema (*Figure 5B,C*). Indeed, *alk5-/-* larvae carrying the rescue transgene exhibited a wild-type morphology and function of the OFT and connecting vessels (*Figure 5D–F*), including a single VA and a wild-type like OFT expansion (*Figure 5G*; *Figure 5—video 1*). Occasionally, the restored cardiac function and blood flow allowed a few (10.2%, n = 108) of the *alk5* rescued mutants to inflate their swim bladder and survive until 9 days post-fertilization (dpf) (*Figure 5—figure supplement 1B,C*). However, despite the vascular rescue, the fish did not survive to adulthood, presumably due to the lack of Alk5 signaling in other tissues, as suggested by an altered morphology of the head (*Figure 5—figure supplement 1C*).

In order to analyze the EC-specific OFT rescue at a cellular level, we performed EdU incorporation experiments and observed that the number of proliferating ECs was reduced in transgenic *alk5-/-* larvae compared with *alk5-/-* not carrying the transgene (*Figure 5H*). Remarkably, the overexpression of *alk5b* in the endothelium of *alk5-/-* larvae also restored the formation of the SMC wall (*Figure 5I–K*). In fact, 75 hpf transgenic *alk5-/-* larvae exhibited SMCs organized in multiple layers around the OFT (*Figure 5K*), and 71.9% of these cells were surrounded by uniform Eln2 immunostaining (*Figure 5—figure supplement 1D*), a percentage similar to that observed in *alk5+/+* OFTs (77.6%). Moreover, the EC-specific overexpression of *alk5b* in *alk5* mutants was sufficient to restore SMC proliferation to wild-type levels (*Figure 5—figure supplement 1E*).

Overall, these data suggest that endothelial Alk5 signaling is sufficient to restore OFT morphology and function, including SMC wall formation, in *alk5* mutants.

## Alk5 signaling regulates ECM gene expression in the cardiac OFT

We hypothesized that Alk5 is required in the OFT and that it controls an expression program which modulates its structural integrity. To identify candidate effector genes, we performed a transcriptomic analysis using manually extracted 56 hpf hearts, including the OFTs, using control and Alk5 inhibitor-treated embryos (*Figure 6—figure supplement 1A*). We chose 56 hpf as the developmental stage for this analysis in order to avoid secondary effects deriving from OFT malfunction.

Analysis of the Alk5 inhibitor-treated samples revealed 955 differentially expressed genes (DEGs), 480 of which were downregulated compared with control (*Supplementary file 1*). Notably, genes downregulated upon Alk5 inhibition encode multiple ECM components and the related Gene Ontology categories are amongst the most enriched (*Figure 6—figure supplement 1B,C*). In order to identify extracellular proteins that might function in signaling between ECs and SMCs, we compared the list of downregulated genes with the secreted factor genes from the zebrafish matrisome (*Nauroy et al., 2018*; *Figure 6—figure supplement 1D*). Amongst the 82 genes identified, five of them were specifically expressed in the OFT in adult zebrafish (*Singh et al., 2016*). Thus, we analyzed the mRNA levels of these five genes by RT-qPCR and observed that they were also decreased in *alk5-/-* compared with *alk5+/+* larval hearts (*Figure 6—figure supplement 1E*). Amongst the candidate genes, *fbln5* was one of the most downregulated in both *alk5-/-* and Alk5 inhibitor-treated larvae (*Figure 6—figure supplement 1F*). Importantly, we observed that *fbln5* expression is highly enriched in the OFT at 72 hpf (*Figure 6A,B*). Fbln5 is an ECM protein that mediates cell-to-matrix adhesion, and it has been reported to serve several functions including the inhibition of EC proliferation *in vitro* and the organization of the elastic lamina (*Sullivan et al., 2007*; *Yanagisawa et al., 2009*). Interestingly, mouse *Fbln5* is expressed in both fibroblasts/SMCs and ECs (*Tabula Muris Consortium et al., 2018*), and also specifically in the embryonic OFT and aorta (*Liu et al., 2019*). To confirm its expression in ECs in zebrafish embryos, we performed *in situ* hybridization at 36 hpf, a stage when SMCs are not yet covering the OFT. We detected *fbln5* expression in the OFT and in the first pair of aortic arches (*Figure 6C,C'*; *Figure 6—figure supplement 2A,A'*), and we further detected its expression in OFT ECs (*Figure 6—figure supplement 2B', B*).

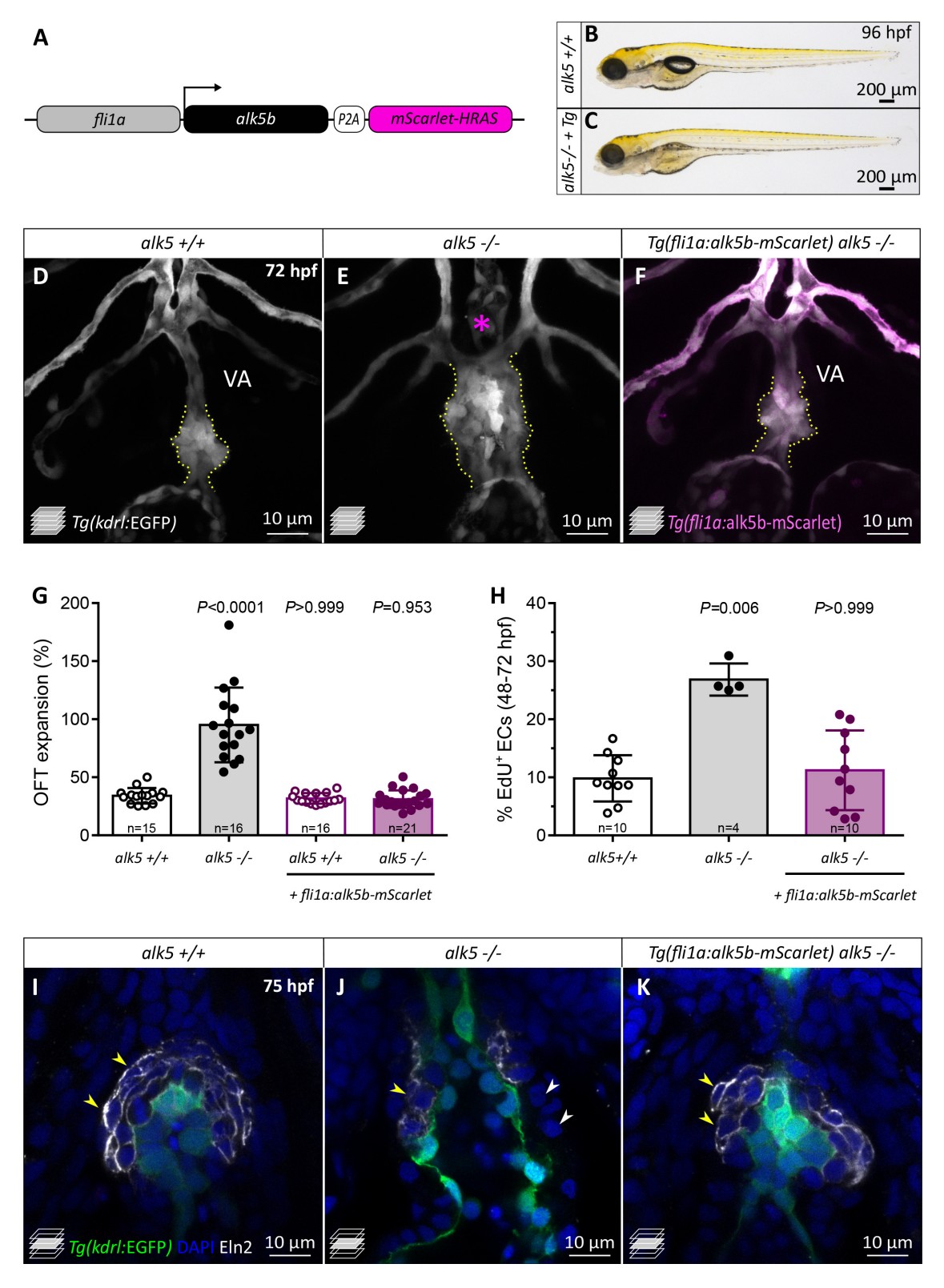

**Figure 5.** *alk5* endothelial-specific overexpression is sufficient to restore OFT wall formation and function in *alk5* mutants. (A) Schematic of the construct used for endothelial-specific rescue experiments. (B, C) Brightfield images of 96 hpf *alk5+/+* and *alk5-/-* larvae carrying the EC-specific *alk5b* rescue transgene (*Tg*). (D–F) Confocal images of the OFT in 72 hpf *Tg(kdrl:EGFP)* larvae, showing the morphological rescue in *Tg(fli1a:alk5b-mScarlet) alk5-/-* animals (F). Asterisk indicates the absence of the VA in *alk5* mutants; dotted lines outline the OFT. (G) Percentage of OFT expansion in 78 hpf *alk5+/+*,

*Figure 5 continued on next page*

Figure 5 continued

*alk5-/-*, and *Tg(fli1a:alk5b-mScarlet) alk5-/-* animals. (H) Percentage of EdU+ ECs in 72 hpf *alk5+/+*, *alk5-/-*, and *Tg(fli1a:alk5b-mScarlet) alk5-/-* animals. (I–K) Confocal images of 75 hpf *Tg(kdrl:EGFP)* larvae immunostained for Eln2. Arrowheads, SMCs devoid of (white) or surrounded by (yellow) Eln2 immunostaining; I, n = 12; J, n = 13, K, n = 17. (G, H) Plot values represent means ± SD; p values from Kruskal-Wallis test, compared with the first column (*alk5+/+*). VA- ventral artery; *Tg- Tg(fli1a:alk5b-mScarlet)*. Scale bars: (B, C) 200 µm; (D–F, I–K) 10 µm. See also ***Figure 5—figure supplement 1***. The online version of this article includes the following video and figure supplement(s) for figure 5:

**Figure supplement 1.** *alk5* endothelial-specific overexpression is sufficient to restore OFT wall formation and function in *alk5* mutants.

**Figure 5—video 1.** Confocal video of the beating heart of a *Tg(kdrl:EGFP) alk5-/-* larva at 78 hpf, carrying the *fli1a:alk5b-mScarlet* transgene and showing OFT expansion.

https://elifesciences.org/articles/57603#fig5video1

The specific expression of *fbln5* in the developing OFT from early developmental stages and its downregulation in *alk5* mutants suggest a potential role for this ECM gene downstream of TGF-β signaling during OFT formation. To test this hypothesis, we analyzed *fbln5* expression in *alk5-/-* hearts when rescued with the endothelial *fli1a:alk5b-mScarlet* transgene. Notably, we observed an increase in *fbln5* mRNA levels by RT-qPCR in the rescued mutant hearts compared with mutant hearts not carrying the transgene, and the extent of this increase seemed to correlate with the level of *alk5b* expression in the rescued animals (***Figure 6—figure supplement 2C***). In order to assess a potential role for Fbln5 downstream of Alk5 in the OFT, we overexpressed *fbln5* globally by injecting *fbln5* mRNA into *alk5* mutants at the one-cell stage. Of note, this approach partially reduced the expansion of the OFT in *alk5-/-* animals, while it did not have an effect in wild types (***Figure 6—figure supplement 2D***). To dig deeper into the potential role of Fbln5 downstream of Alk5 in ECs, we generated a transgenic line overexpressing *fbln5* in the endothelium (*Tg(fli1a:fbln5)*) in an *alk5-/-* background. Notably, the EC-specific overexpression of *fbln5* was sufficient to partially rescue OFT morphology and expansion in *alk5-/-* larvae (***Figure 6D–G***), as well as VA formation.

When analyzing the cell-specific defects, we observed that *alk5* mutants injected with *fbln5* mRNA exhibited a percentage of EdU+ ECs (16.0%), comparable with wild types (15.6%) at early embryonic stages (***Figure 6—figure supplement 2E***). Additionally, since *fbln5* has been shown to stabilize the elastic lamina in other contexts (***Nakamura et al., 2002***; ***Chapman et al., 2010***), we aimed to assess its role specifically in organizing the OFT ECM. Importantly, EC-specific *fbln5* overexpression in *alk5* mutants led to a better organization of Eln2 immunostaining in the OFT, surrounding 58.5% of SMCs, compared with 31.1% in *alk5-/-* animals not carrying the transgene (***Figure 6H–K***). This level of rescue was also achieved by *fbln5* mRNA injections into one-cell stage embryos (56.4%; ***Figure 6—figure supplement 2F***), suggesting that a great extent of the role of Fbln5 in the OFT can be carried out by its endothelial expression.

Overall, these data suggest that Fbln5 plays an important role in regulating the OFT ECM downstream of Alk5 signaling, leading to increased elastin organization and restricting EC proliferation.

## Discussion

Taking advantage of the zebrafish model, we report an important role for the TGF-β receptor I Alk5 in cardiac OFT morphogenesis. We show that TGF-β signaling through Alk5 restricts EC proliferation during early embryonic stages. Furthermore, loss of Alk5 causes defects in the SMC wall of the OFT. The combination of these phenotypes leads to defects in OFT structural integrity, eventually resulting in vessel dissection, reminiscent of human pathologies. Notably, restoring Alk5 expression in the endothelium is sufficient to rescue the OFT defects, suggesting a critical role for Alk5 in ECs.

### Cell-specific role of Alk5 and TGF-β signaling in the cardiac OFT development

The TGF-β pathway's multi-functional role results in many context- and tissue-dependent functions, which are difficult to unravel (***Pardali et al., 2010***). Most studies on diseases of the great arteries have aimed to understand the role of TGF-β signaling in SMCs (***Li et al., 2014***; ***Yang et al., 2016***; ***Zhang et al., 2016***; ***Perrucci et al., 2017***; ***Takeda et al., 2018***), overlooking its function in the endothelium. In EC biology, Alk5 function has been mostly studied *in vitro*, where it has been shown to maintain EC quiescence (***Goumans et al., 2002***; ***Goumans et al., 2003***; ***Lebrin et al., 2004***;

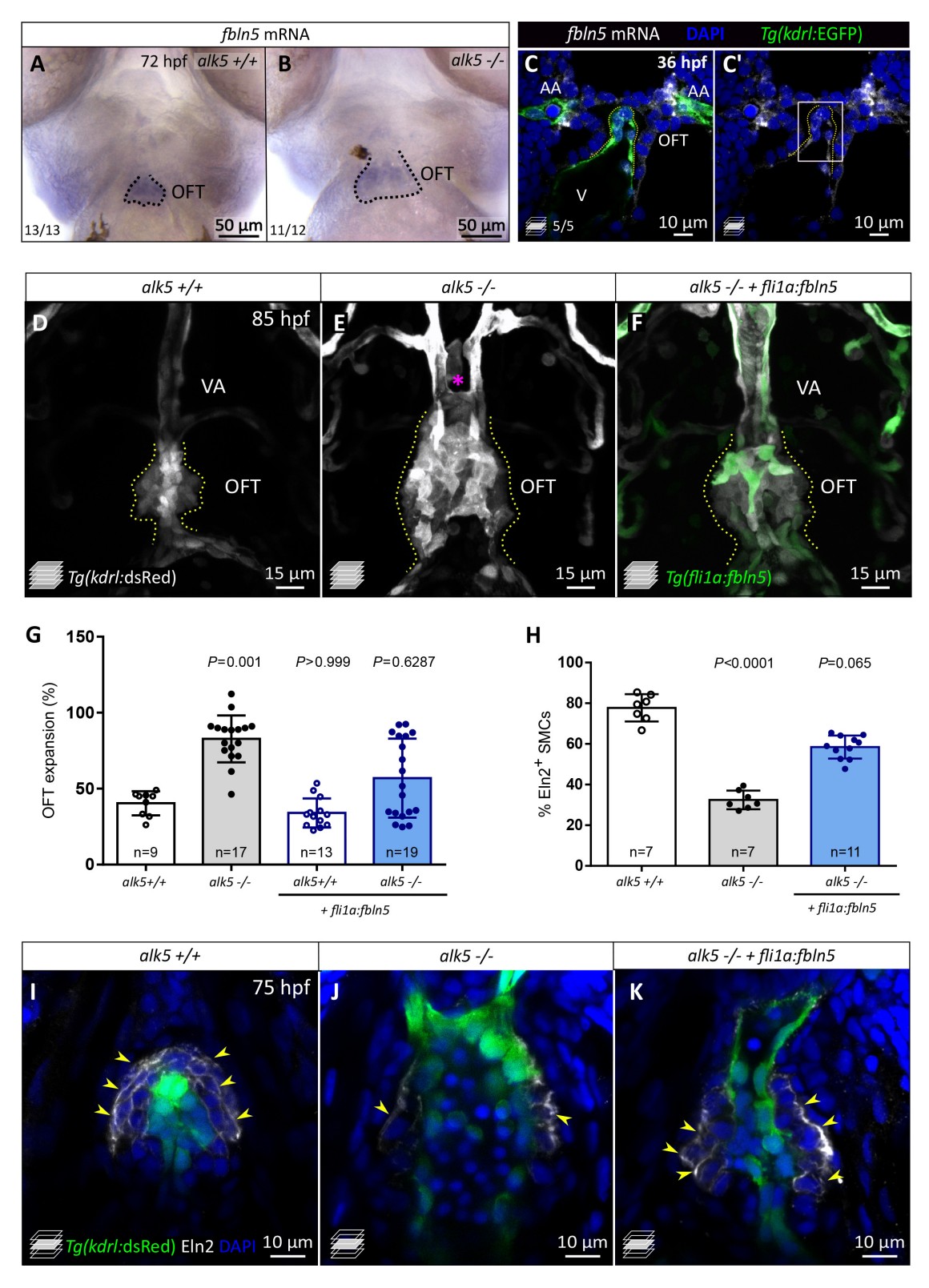

**Figure 6.** *fbln5* endothelial-specific expression partially rescues the cardiac OFT defects in *alk5* mutants. (**A, B**) Whole-mount *in situ* hybridization for *fbln5* expression in the OFT of 72 hpf *alk5+/+* (n = 13) and *alk5-/-* (n = 12) larvae. Dotted lines outline the OFT. (**C, C'**) Fluorescent *in situ* hybridization for *fbln5* expression (white) in the OFT of 36 hpf wild-type embryos. Green, ECs. Dotted lines outline the OFT, boxed area is shown in ***Figure 6—figure supplement 2B,B'***. (**D–F**) Confocal images of 85 hpf *Tg(kdrl:dsRed) alk5+/+* (n = 7), *alk5-/-* (n = 9), *Tg(fli1:fbln5) alk5-/-* (n = 9) larvae, showing the partial

*Figure 6 continued on next page*

*Figure 6 continued*

morphological rescue of the OFT and VA in *Tg(fli1:fbln5) alk5* -/- larvae (6/9, **F**). Asterisk indicates the absence of the VA in *alk5* mutants; dotted lines outline the OFT. (**G**) Percentage of OFT expansion in 78 hpf *alk5+/+*, *alk5-/-*, and *Tg(fli1:fbln5) alk5-/-* animals. (**H**) Percentage of SMCs surrounded by Eln2 immunostaining (per sagittal plane) at 75 hpf. (**I–K**) Confocal images of 75 hpf larvae immunostained for Eln2. Yellow arrowheads, SMCs surrounded by Eln2 immunostaining; I, n = 7; J, n = 7; K, n = 11. (**G, H**) Plot values represent means ± SD; p values from Kruskal-Wallis tests, compared with the first column (*alk5+/+*). AA- aortic arch, VA- ventral artery. Scale bars: (**A, B**) 50 µm; (**C, C'**, I–K) 10 µm; (**D–F**) 15 µm. See also *Figure 6—figure supplements 1–2*.

The online version of this article includes the following figure supplement(s) for figure 6:

**Figure supplement 1.** Alk5 regulates the expression of ECM genes in the cardiac OFT, including that of *fbln5*.

**Figure supplement 2.** *fbln5* is expressed in the cardiac outflow tract endothelium and its global overexpression partially rescues OFT defects in *alk5* mutants.

*van Meeteren and ten Dijke, 2012*; *Maring et al., 2016*). In the zebrafish model, we confirmed a pivotal role for Alk5 in restricting EC proliferation *in vivo*. We found that ECs lacking Alk5 function also exhibited impaired movement in the OFT. Interestingly, *alk5* mutants fail to form the ventral aorta. Further studies will be needed to resolve the mechanisms responsible for VA formation and its impairment in *alk5* mutants. Moreover, it will be interesting to analyze in detail the aortic arch patterning in *Alk5-/-* mice, potentially unveiling defects in the connection between the OFT and the pharyngeal vessels.

Importantly, the early EC defects together with the endothelial-specific rescue data suggest that ECs are primarily responsible for the OFT phenotype in *alk5-/-* zebrafish. In fact, despite a broad *alk5/Alk5* receptor expression in the zebrafish heart, the signaling appears to be initially enriched in the OFT endothelium. Similarly, while *Alk5* expression in mouse appears to be enriched in the aortic SMCs (*Seki et al., 2006*), a few studies have suggested a pivotal role for TGF-β signaling in ECs (*Sridurongrit et al., 2008*; *Bochenek et al., 2020*). For example, although *Tgfbr2* deletion in SMCs leads to OFT expansion in mouse (*Jaffe et al., 2012*), only EC-specific *Alk5* KO mice display the early cardiovascular defects of mice carrying a global *Alk5* mutation (*Carvalho et al., 2007*; *Sridurongrit et al., 2008*; *Dave et al., 2018*). This discrepancy indicates distinct requirements for various TGF-β receptors in different cell types in the OFT, possibly at different developmental stages.

Remarkably, both mice and patients exhibiting aneurysms display a hyperactivation of down-stream TGF-β signaling, despite carrying loss-of-function mutations in TGF-β genes (*Jones et al., 2009*; *Mallat et al., 2017*; *Takeda et al., 2018*). In zebrafish *alk5* mutants, we initially observed a decrease in p-Smad3 signal before the appearance of the phenotype, and an exacerbation of the defect in *alk5-/-* embryos treated with a Smad3-specific inhibitor. In contrast, we then observed increased activation of the TGF-β pathway at later stages (75 hpf, data not shown), when –as in patients- the phenotype is already evident. These data suggest that the initial cause of the pheno-type could arise from an early downregulation of the TGF-β pathway in ECs, providing new insights into a complex signaling event.

## Endothelium-smooth muscle interplay in the cardiac OFT and aorta

Surprisingly, we found that *alk5* overexpression in ECs was sufficient to restore SMC wall formation in *alk5* mutants. Although these results do not resolve whether ECs are the only cell type in which Alk5 is necessary for OFT morphogenesis, together with the initial activation of the TGF-β pathway in OFT ECs, they suggest that the SMC phenotype is a secondary effect from an endothelial defect. Perturbed interactions between ECs and SMCs have been implicated in different human pathologies including pulmonary hypertension, atherosclerosis and arteriovenous malformations (e.g. hereditary hemorrhagic telangiectasia) (*Mancini et al., 2009*; *Gao et al., 2016*; *Cunha et al., 2017*; *Li et al., 2018*).

The communication between ECs and SMCs can occur in different ways including via physical contact, exchange of signaling cues, and ECM deposition (*Gaengel et al., 2009*; *Lilly, 2014*; *Li et al., 2018*; *Sweeney and Foldes, 2018*), all of which are likely to play a role in the *alk5* mutant phenotype. In addition, during development, SMCs are recruited to the vessels once the initial establishment of the endothelial layer is complete (*Stratman et al., 2017*; *Sweeney and Foldes, 2018*). Thus, one can speculate that enhanced EC proliferation and remodeling in *alk5* mutants

represent a less mature state of the vessel, thereby inhibiting SMC coverage of the OFT. Later on, reduced SMC coverage might further drive EC hyperproliferation in a positive feedback loop, contributing to the severity of the phenotype.

Overall, we suggest that the severe SMC phenotype causing the functional OFT defects and leading to aortic aneurysms in patients might be masking the important role of the endothelium in the early etiology of these pathologies. Therefore, it will be important to further characterize the role of ECs in SMC stabilization and vascular wall integrity.

## Identifying new molecular regulators of cardiac OFT development and disease

Analyses at several cellular and molecular levels provided insights into the structural changes directly linked to the functional phenotype in *alk5* mutants. Together with the observed defective elastic lamina and altered intercellular space, the transcriptomic data identified differential expression of several ECM component genes following inhibition of Alk5 function. The ECM is secreted by both ECs and SMCs and is a source of signaling mediators crucial for their interaction (*Kelleher et al., 2004*; *Davis and Senger, 2005*). One of the most promising ECM genes downregulated after Alk5 inhibition is *fbln5,* which encodes an integrin-binding extracellular protein (*Nakamura et al., 1999*). In mouse, *Fbln5* is expressed, and its protein secreted, by both ECs and fibroblasts or SMCs (*Nakamura et al., 1999*; *Tabula Muris Consortium et al., 2018*; *Liu et al., 2019*) and, in zebrafish, we found that it serves as a very specific marker for the OFT, starting from early developmental stages. Moreover, *FBLN5* expression has been shown to be directly induced by TGF-β signaling *in vitro* (*Albig and Schiemann, 2004*; *Kuang et al., 2006*; *Topalovski et al., 2016*). FBLN5 plays multiple functions such as assembling the elastic lamina surrounding SMCs (*Nakamura et al., 2002*; *Chapman et al., 2010*) and promoting EC-to-ECM attachment *in vitro* (*Preis et al., 2006*; *Williamson et al., 2007*). The adhesion of ECs to the matrix, which is mediated in part by FBLN5, appears to be essential to restrict their proliferation, suggesting that FBLN5 functions as an anti-angiogenic factor (*Albig and Schiemann, 2004*; *Sullivan et al., 2007*). By enhancing the levels of *fbln5* in *alk5* mutants both globally and specifically in the endothelium, we were able to partially restore OFT morphology and function, as well as Eln2 coverage of SMCs. Although *fbln5* is expressed in both ECs and SMCs, its overexpression specifically in the endothelium seemed sufficient to ameliorate the *alk5* mutant phenotype. It is thus conceivable that ECs initiate the generation of a local extracellular environment around the OFT, which later is maintained and refined with the combined secretion of ECM components by both cell types.

Given the prevalence of OFT-related cardiovascular diseases, there is a need for multiple model systems that allow one to investigate the causes of these pathologies at the cellular and molecular levels, and the zebrafish could complement the mouse in this regard. Furthermore, the overlap of our transcriptomic data with the genes associated with aortic aneurysm (*Brownstein et al., 2017*; *Kim and Stansfield, 2017*), and the similarities between the endothelial ruptures in *alk5* mutants and those in human aortic dissection suggest that the zebrafish could serve as a valuable model for aneurysm research.

## Materials and methods

**Key resources table**

| Reagent type (species) or resource | Designation | Source or reference | Identifiers | Additional information |
|---|---|---|---|---|
| Genetic reagent (*Danio rerio*) | *Tg(kdrl:EGFP)s843* | *Jin et al., 2005* | ZFIN: *s843* | |
| Genetic reagent (*D. rerio*) | *Tg(kdrl: TagBFP)mu293* | *Matsuoka et al., 2016* | ZFIN: *mu293* | |

*Continued on next page*

*Continued*

| Reagent type (species) or resource | Designation | Source or reference | Identifiers | Additional information |
|---|---|---|---|---|
| Genetic reagent (D. rerio) | Tg(kdrl: dsRed2)pd27 | *Kikuchi et al., 2011* | ZFIN: *pd27* | |
| Genetic reagent (D. rerio) | Tg(fli1a:Gaf4ff)ubs4 | *Zygmunt et al., 2011* | ZFIN: *ubs4* | |
| Genetic reagent (D. rerio) | Tg(UAS:Kaede)rk8 | *Herwig et al., 2011* | ZFIN: *rk8* | |
| Genetic reagent (D. rerio) | TgBAC(pdgfrb: EGFP)ncv22 | *Ando et al., 2016* | ZFIN: *ncv22* | |
| Genetic reagent (D. rerio) | Tg(kdrl:NLS-mCherry)is4 | *Wang et al., 2010* | ZFIN: *is4* | |
| Genetic reagent (D. rerio) | Tg(12xSBE: EGFP)ia16 | *Casari et al., 2014* | ZFIN: *ia16* | |
| Genetic reagent (D. rerio) | TgBAC(tgfbr1b: EGFP,cryaa: CFP)bns330 | This manuscript | ZFIN: *bns330* | |
| Genetic reagent (D. rerio) | Tg(fli1a:tgfbr1b-P2A-mScarlet-Hsa.HRAS)bns421 | This manuscript | ZFIN: *bns421* | |
| Genetic reagent (D. rerio) | tgfbr1a$^{bns329}$ | This manuscript | ZFIN: *bns329* | |
| Genetic reagent (D. rerio) | tgfbr1b$^{bns225}$ | This manuscript | ZFIN: *bns225* | |
| Antibody | Anti-GFP (chicken polyclonal) | AvesLab | Cat#: GFP-1020 | 1:400 |
| Antibody | Anti-Elastin2 (rabbit polyclonal) | *Miao et al., 2007* | | 1:100 |
| Antibody | Anti-tRFP (rabbit polyclonal) | Evrogen | Cat# AB233 | 1:200 |
| Antibody | Secondaries Alexa FluorTM 488-568-647 IgG (H+L) (goat polyclonal) | Thermo Fisher Scientific | | 1:500 |
| Antibody | Anti-Smad3 (phospho S423 + S425) antibody (rabbit monoclonal) | Abcam | Cat# ab52903 | 1:100 |
| Antibody | nti-phospho-Smad1/5 (Ser463/465)/Smad9 (Ser465/467) (rabbit monoclonal) | Cell Signaling Technology | Cat# 13820 | 1:100 |
| Antibody | Anti-dsRed (rabbit polyclonal) | Takara Bio Clontech | 632496 | 1:200 |
| Antibody | Anti-Alcama/Dm-Grasp (mouse monoclonal) | DSHB | ZN-8 | 1:50 |

*Continued on next page*

*Continued*

| Reagent type (species) or resource | Designation | Source or reference | Identifiers | Additional information |
|---|---|---|---|---|
| Commercial assay or kit | Click-iT EdU Cell Proliferation Kit for Imaging, Alexa Fluor 647 dye | Thermo Fisher Scientific | Cat# C10340 | |
| Commercial assay or kit | In-Fusion HD Cloning Plus | Takara Bio | Cat# 638910 | |
| Commercial assay or kit | DyNAmo ColorFlash SYBR Green qPCR Mix | Thermo Scientific | Cat# F416S | |
| Chemical compound, drug | E-616452 | Cayman Chemical | 14794 | |
| Chemical compound, drug | SIS3 | Calbiochem/ Merck | 566405 | |
| Chemical compound, drug | EdU | Thermo Fisher Scientific | Cat# A10044 | |
| Chemical compound, drug | FITC-dextran 2000 kDa | Sigma | Cat# 52471 | |
| Chemical compound, drug | LysoTracker Deep Red | Thermo Fisher Scientific | Cat# L12492 | |
| Commercial assay or kit | miRNeasy micro Kit | Qiagen | Cat# 217084 | |
| Commercial assay or kit | mMessage mMachine T3 Transcription Kit | Thermo Fisher Scientific | Cat# AM1348 | |
| Commercial assay or kit | mMessage mMachine T7 Transcription Kit | Thermo Fisher Scientific | Cat# AM1344 | |
| Commercial assay or kit | DIG RNA labeling kit | Roche | Cat# 11277073910 | |
| Commercial assay or kit | Maxima First Strand cDNA kit | Thermo Fisher Scientific | Cat# K1641 | |
| Commercial assay or kit | MegaShortScript T7 | Thermo Fisher Scientific | Cat# AM1354 | |
| Commercial assay or kit | RNA Clean and Concentrator Kit | Zymo Research | Cat# R1013 | |
| Software, algorithm | ZEN Blue 2012 | Zeiss, Germany | | |
| Software, algorithm | ZEN Black 2012 | Zeiss, Germany | | |
| Software, algorithm | Imaris - Version 8.4.0 | Bitplane, UK | | |
| Software, algorithm | GraphPad Prism 6 | GraphPad Software, USA | | |
| Software, algorithm | FIJI/ImageJ | *Schindelin et al., 2012* | | |

## Zebrafish husbandry and lines

Larvae were raised under standard conditions. Adult fish were maintained in 3.5 l tanks at a stock density of 10 fish/l with the following parameters: water temperature: 27–27.5°C; light:dark cycle: 14:10; pH: 7.0–7.5; Conductivity: 750–800 μS/cm. Fish were fed three to five times a day, depending on age, with granular and live feed (*Artemia salina*). Health monitoring was performed at least once a year. Embryos were grown at 28°C and staged at 50–75% epiboly to determine synchronization.

The following lines were used in the study:

*Tg(kdrl:EGFP)s843* (*Jin et al., 2005*); *Tg(kdrl:TagBFP)mu293* (*Matsuoka et al., 2016*); *Tg(kdrl: dsRed2)pd27* (*Kikuchi et al., 2011*); *Tg(fli1a:Gaf4ff)ubs4* (*Zygmunt et al., 2011*) in combination with *Tg(UAS:Kaede)rk8* (*Herwig et al., 2011*), abbreviated *Tg(fli1a:Kaede)*; *TgBAC(pdgfrb:EGFP)ncv22* (*Ando et al., 2016*), abbreviated *Tg(pdgfrb:EGFP)*; *Tg(12xSBE:EGFP)ia16* (*Casari et al., 2014*), *Tg (kdrl:NLS-mCherry)is4* (*Wang et al., 2010*).

## Generation of transgenic lines

To generate *TgBAC(tgfbr1b:EGFP,cryaa:CFP)bns330*, abbreviated *TgBAC(alk5b:EGFP)*, we used the BAC clone CH1073-59L16 (BACPAC Resources Center), containing 37.3 kb of the *tgfbr1b* locus. All recombineering steps were performed as previously described (*Bussmann and Schulte-Merker, 2011*), except for the addition of a cryaa:CFP-iTol2 cassette to identify transgenic animals. In addition, we removed the kanamycin-resistance cassette in the last step, taking advantage of Flippase recombination. The following homology arms were used to amplify and recombine the EGFP_Kan cassette: *alk5b* HA1 GFP fw 5'- ACTGGAGTCCAGCAGGAGAACAGAAGAGGAGCGGGATTATC TCCAGGAGGACCATGGTGAGCAAGGGCGAGGAG-3', *alk5b* HA2 Kan rv 5'- ACCAGTCGA-CACGGCACTGCCTCCATCATCATCATCTTCATCATCATCTTTTCCAGAAGTAGTGAGGAG −3'. The following homology arms were used to amplify and recombine the cryaa:CFP_iTol2 cassette: Fosmid iTol2_eye fw: 5'-TTCTCTGTTTTTGTCCGTGGAATGAACAATGGAAGTCCGAGCTCATCGCTCCC TGCTCGAGCCGGGCCCAAGTG −3'; Fosmid iTol2_eye rv: 5'- AGCCCCGACACCCGCCAA-CACCCGCTGACGCGAACCCCTTGCGGCCGCATGAAACAGCTATGACCATGTAA-3'.

To generate *Tg(fli1a:tgfbr1b-P2A-mScarlet-Hsa.HRAS)bns421*, abbreviated *Tg(fli1a:alk5b-mScarlet)*, the full-length *tgfbr1b* (NM_001115059) coding sequence was amplified using the primers *alk5b* ATG fw 5'- GGATGAAGATGATGATGAAGATG −3' and *alk5b* noSTOP rev 5'- CATCTTGATTCCC TCCTGct −3'. *alk5b* was cloned downstream of the *fli1a* enhancer-promoter sequence (*fli1ep*) (*Villefranc et al., 2007*), into a plasmid containing the P2A sequence (*Kim et al., 2011*) and the mScarlet coding sequence, fused with Hsa.HRAS human sequence for membrane localization. The *fli1ep* sequence was amplified with the following primers: *fli1a* fw 5'-TTGGAGATCTCATCTTTGAC-3', *fli1a* rev 5'-GGTGGCGCTAGCTTCGCGTCTGAATTAATTCC-3'.

To generate *Tg(fli1a:fbln5,EGFP)*, abbreviated *Tg(fli1a:fbln5)*, the full-length *fbln5* (NM_001005979) coding sequence was amplified by wild-type cDNA with the following primers *fbln5* ATG fw 5'-ATGTTTGTTGAACTACGTGGC-3', *fbln5* STOP rv 5'-TCAGAATGGATGTTCGGA-GAC-3'. The *fbln5* CDS was cloned in the *fli1aep:EGFP,mCherry* (*Nicoli et al., 2010*) substituting the mCherry CDS. The experiments were conducted on F1 embryos and larvae deriving from the same F0 founder.

All cloning experiments, except for the BAC, were performed using InFusion Cloning. All vectors had *Tol2* elements to facilitate genome integration.

## Generation of mutant lines using CRISPR/Cas9 technology

In order to generate *tgfbr1a^bns329^* (referred to as *alk5a* mutants) and *tgfbr1b^bns225^* (referred to as *alk5b* mutants) mutant lines, a CRISPR design tool (https://chopchop.cbu.uib.no/) was used to design sgRNAs. For *tgfbr1a ^bns329^*, a sgRNA targeting exon 1 (targeting sequence: GGATGATC TTTACCCCC) and for *tgfbr1b^bns225^*, a sgRNA targeting exon 4 (targeting sequence: GAGGAGCGC TCCTGGTTC) were assembled as described previously (*Gagnon et al., 2014*; *Varshney et al., 2015*). Briefly, sgRNAs and *Cas9* mRNA were transcribed *in vitro* with a MegaShortScript T7 Tran-scription Kit and an mMESSAGE mMACHINE T3 Transcription Kit, respectively, and purified with RNA Clean and Concentrator Kit. To synthesize *Cas9* mRNA, pT3TS-nCas9n (Addgene) was used as a template. 50 pg sgRNA and 300 pg *Cas9* mRNA were injected into 1 cell stage embryos. The mutants were genotyped using high-resolution melt analysis (HRMA) (Eco-Illumina) with the

following primers: *alk5a* HRM fw: 5'- CTTCTGGACAGACCGTGACA-3', *alk5a* HRM rv: 5'- GAAG-GAGCGCACTGGAAAG −3', *alk5b* HRM fw: 5'- CGCTGGAGAGGAGAGGAGG −3', *alk5b* HRM rv: 5'- GTCTCAGCATGACGGTCTGG −3', and/or by sequencing; *alk5* mutants carrying the *Tg(fli1a: alk5b-mScarlet)* transgene were genotyped by HRMA with the following primers, in order to amplify exclusively the genomic locus: *alk5b* HRM rescue fw: 5'- AGAGGAGAGGGAGGTGGCG-3', *alk5b* HRM intron rv: 5'- CCCATGATGCCCCAGTGC −3'.

The *tgfbr1a* [bns329] fish carry a 2 bp insertion and a 6 bp deletion in exon 1, while the *tgfbr1b* [bns225] fish carry a 3 bp insertion and an 11 bp deletion in exon 4. *alk5a+/-; alk5b+/-* fish were incrossed to obtain *alk5a-/-;alk5b+/-* and *alk5a+/+;alk5b+/+* adults. For the majority of the experiments, *alk5-/-* (*alk5a-/-;alk5b-/-*) embryos and larvae were obtained from *alk5a-/-;alk5b+/-* parents, while *alk5+/+* embryos and larvae were obtained from *alk5a+/+;alk5b+/+* parents; thus, *alk5-/- and alk5+/+* animals were first generation cousins. Larvae analyzed in *Figure 1—figure supplement 1M* and *Figure 4A–C* were obtained from *alk5a+/-; alk5b+/-* parents; thus, *alk5+/+ and alk5-/-* animals are siblings.

Injection of *fbln5* mRNA *fbln5* (NM_001005979) complete coding sequence was amplified by wild-type cDNA with the following primers *fbln5* ATG fw 5'-ATGTTTGTTGAACTACGTGGC-3', *fbln5* STOP rv 5'-TCAGAATGGATGTTCGGAGAC-3' and cloned into the pCS2⁺ plasmid. The mRNA was synthesized *in vitro* using the mMESSAGE mMACHINE T7 Transcription Kit and purified with RNA Clean and Concentrator. 150 pg of *fbln5* mRNA was injected into one-cell stage embryos deriving from *alk5a-/-; alk5b+/-* and *alk5+/+* crosses, and non-injected embryos deriving from the same crosses were used as controls.

## Dextran injections

Larvae at 96 hpf were embedded in low-melting agarose containing 0.01% Tricaine (Sigma) and positioned on their side. They were injected with 2 nl of fluorescein isothiocyanate (FITC)-dextran 2000 kDa in the portion of the posterior cardinal vein on top of the yolk elongation. After verifying the correct injection on a Nikon SMZ25 stereomicroscope, larvae were recovered for 10 min in egg water and then embedded in 1% low melting agarose containing 0.2% tricaine. The larvae were then imaged with a confocal microscope and subsequently genotyped.

## Chemical treatments

The Alk5 inhibitor II, E-616452 (*Gellibert et al., 2004*), was dissolved in DMSO to create a stock solution (10 mM) and frozen in aliquots for single use. Embryos were treated with the inhibitor, at a final concentration of 2.5 µM (diluted in egg water), starting at 36 hpf until the desired stage for analysis. Different concentrations were previously tested to minimize off-target developmental defects and, for the desired stage, 2.5 µM appeared to be the lowest effective dose. Control embryos were treated for the same time with 0.025% DMSO.

For Smad3 inhibitor experiments, SIS3 (*Jinnin et al., 2006*; *Dogra et al., 2017*) was dissolved in DMSO to create a 5 mM stock solution and frozen in aliquots for single use. Embryos were treated with the inhibitor, at a final concentration of 1.5 µM (diluted in egg water), or DMSO from 36 hpf until 75 hpf, protected from light.

For LysoTracker experiments, 95 hpf larvae were incubated at 28°C for 1 hr in egg water containing 10 µM LysoTracker Deep Red (Thermo Fisher Scientific) and rinsed several times with fresh water before imaging by confocal microscopy.

## Whole mount immunostaining

Whole mount immunostaining was performed as described previously (*Gunawan et al., 2019*), with some modifications. Briefly, embryos or larvae were fixed in 4% PFA after stopping the heart with 0.2% Tricaine to prevent the collapse of the tissues during fixation. Embryos were then washed and deyolked using forceps. The blocking step was performed in PBS/1% BSA/1%DMSO/0.5% Triton X-100 (PBDT) and 5% goat serum. Primary and secondary antibody incubations were performed overnight at 4°C. Embryos were incubated with 1 µg/ml DAPI during secondary antibody incubation.

Primary antibodies used were GFP (1:400), Elastin2 (1:100), tRFP (1:200), anti-dsRed (1:200), anti-Alcama/Dm-Grasp (1:50), anti-p-Smad3 (1:100) and anti-pSmad1/5/8 (1:100). Secondary antibodies were used at 1:500 concentration.

Elastin2 antibody was purified from the previously described serum stock (*Miao et al., 2007*). For all the immunostainings, animals were genotyped after the required image analyses.

## Whole mount *in situ* hybridization

The following primers were used to generate the DNA template for *in situ* RNA probes: *alk5b* ISH fw: 5'- TCATGAGCACGGTTCTCTGT - 3', *alk5b* ISH T7 rv: 5'- GTAATACGACTCACTATAGGA TCCCCAATGAGTCGGTGTT - 3', *fbln5* ISH fw: 5'- TGTTCTCACAGGTTCTTGCC- 3', *fbln5* ISH T7 rv: 5'- TAATACGACTCACTATAGGGCAGGGCTACAGGAACAGGAA- 3'.

Digoxigenin (DIG)-labeled probes were transcribed *in vitro*, with T7 polymerase (Promega) and DIG RNA labeling kit (Roche), using as template the PCR amplicon. Larvae were then collected at the desired stage and fixed with 4% PFA overnight at 4°C. *In situ* hybridization was performed as described previously (*Thisse and Thisse, 2008*). For *Figure 6A,B* and *Figure 6—figure supplement 2A,A'*, anti-GFP primary antibody (1:300 dilution) labeling the endothelium of *Tg(kdrl:EGFP)* larvae was added together with the anti-Dig antibody. The secondary antibody anti-chicken Alexa Fluor 488 (Thermo Fisher Scientific) was added after BM Purple treatment, for 2 hr at room temperature, and followed by PBS/Tween 0.1% (PBST) washes before imaging. For fluorescent *in situ* hybridization, the same protocol of the whole mount bright-field *in situ* was conducted with the following modifications. During the first day, embryos were incubated in 1% $H_2O_2$ for 30 min. During the second day, after the blocking step, embryos were incubated with the primary antibodies anti-DIG-POD (Roche, 1:500) and anti-dsRed (1:200). The next day, after six washes in PBST, embryos were incubated with Amplification solution (TSA fluorescein detection kit) for 2 min. FITC (1:100) was added to the Amplification solution to develop the signal. The embryos were then incubated in the mix for 30 min in the dark at RT and washed in PBST. Embryos were then incubated 2 hr with secondary antibody solution (goat anti-rabbit 568, 1:500) and 1 μg/ml DAPI, and then extensively washed in PBST before imaging.

Embryos were imaged on a Nikon SMZ25 stereomicroscope with a 2×/0.3 objective or LSM700 Axio Imager 40x objective and, if necessary, genotyped afterward.

## EdU assays

Embryos were incubated in egg water containing 0.5 mM EdU/0.5% DMSO (12 animals/4 ml in 6-well plates). Embryos were treated from 24 to 36 hpf or 48 to 72 hpf, according to the experiment, followed by fixation and whole-mount immunostaining, with anti-GFP, anti-Eln2, and DAPI, as described above. The Click-iT reaction was performed after the incubation with the secondary antibody, following the Click-iT EdU Cell Proliferation Kit for Imaging, Alexa Fluor 647 dye manufacturer protocol.

The percentage of EdU$^+$ ECs was calculated with the following formula: % EdU$^+$ cells = (number of DAPI$^+$GFP$^+$ EdU$^+$ cells/total number of DAPI$^+$GFP$^+$ cells in the (OFT) x 100). Differently, for *Figure 4D* and *Figure 5—figure supplement 1E*, the percentage of EdU$^+$ cells refers to the number of DAPI$^+$GFP$^-$ cells (SMCs) in the region of the OFT.

As we observed different percentages of labeled cells depending on the batch of EdU, all the replicates and internal controls for each experiment were performed using the same batch of EdU.

## Confocal microscopy imaging

For live confocal imaging, embryos or larvae were embedded in 1% low-melting agarose/egg water with 0.2% or 0.01% tricaine to image stopped or beating hearts, respectively. Fixed embryos were mounted in 1% low-melting agarose/egg water without tricaine. All embryos or larvae were imaged with LSM700 Axio Imager, LSM800 Axio Examiner or LSM880 Axio Examiner confocal microscope with a W Plan-Apochromat 40×/1.0 or W Plan-Apochromat 20×/1.0 dipping lenses, unless otherwise stated. In order to image the heart region, embryos from 54 hpf onwards were imaged in ventral view. In *Figure 1—figure supplement 1A–B', M*, embryos were mounted on their side to visualize the trunk.

Embryos at 36 hpf were fixed with 4% PFA overnight at 4°C. Embryos were then washed with PBS/0.1% Tween20. The yolk and most of the yolk sac of the embryos were then removed using forceps. Before mounting, the trunk was removed and the heads of the embryos were mounted with the ventral side toward the objective. Therefore, the images were taken accessing to the heart from

the region where the yolk was previously located. The same mounting strategy was used to image all the embryos at 36 hpf after EdU assays.

To realize videos of beating hearts, fish were mounted in 0.8% low-melting agarose/egg water, containing 0.01% tricaine or without tricaine for heart rate measurements. Videos were acquired with an exposure time of 10 ms, at 100 frames/s for 150 frames, covering approximately 3–4 heart-beats; for heart rate measurements, videos were acquired with 5 ms exposure for 5000 frames (approximately 30 s). Videos were acquired with an LD C-Apochromat 40×/1.1 water immersion lens (Zeiss Cell Observer spinning disk microscope with a CSU-X1 scanner unit [Yokogawa] and Hama-matsu ORCAflash4.0 sCMOS cameras).

Time-lapse videos were acquired with LSM800 Axio Observer microscope with Plan-Apochromat 20X/0.8. Embryos were mounted in 0.8% low-melting agarose/egg water with 0.01% tricaine in four-well Chambered Coverglass (Thermo Fisher Scientific) and covered with egg-water containing DMSO or Alk5 inhibitor solution, as described above. Stacks of the OFT and connecting vessels were acquired every 45 min for 24 cycles (18 hr) starting from 56 hpf, after the photoconversion. Only fish that survived until the end of the time-lapse were subsequently analyzed.

All images were acquired using the Zeiss ZEN program.

## Photoconversion

*Tg(fli1a:GAL4FF); Tg(UAS:Kaede)* (*Hatta et al., 2006*) embryos were treated with Alk5 inhibitor or DMSO as described above and embedded in 1% low-melt agarose with 0.2% tricaine at 54 hpf. Pho-toconversion was performed on a LSM880 Axio Examiner confocal microscope with a W Plan-Apo-chromat 20×/1.0 dipping lens. Embryos, mounted ventrally to expose the OFT, were first imaged with 488 and 561 nm lasers to establish a region of interest (ROI) and verify the absence of red signal before photoconversion. The ROI was photoconverted with three scans (4.5 s) of the 405 nm laser. Then, another scan with only 488 and 561 nm lasers was performed to verify the efficiency of the photoconversion. Embryos were quickly removed from the agarose, incubated at 28°C in separate wells of a 24-well plate in the dark to recover the heartbeat. According to the experiment, embryos were kept in the incubator until 56 or 74 hpf. 74 hpf larvae (*Figure 3J–K'*; *Figure 3—figure supplement 1I*) were imaged with an LSM880 Axio Examiner to locate the photoconverted cells, while 56 hpf embryos were imaged overnight with time-lapses as described above (*Figure 3—videos 1* and *2*).

## Defining landmarks for OFT analyses

Since markers to label the zebrafish OFT are currently lacking, we defined it using anatomical fea-tures, similar to what was previously described (*Sidhwani et al., 2019*). The proximal boundary was defined by the point of endocardium and myocardium constriction between the ventricle and the OFT. From 54 hpf onwards for the photoconversion experiments, this constriction in the endocar-dium has also been named the bulbo-ventricular (BV) canal, since it represents the region where the BV valve will form. The distal boundary was determined, up to 56 hpf, by the point of bifurcation of the OFT endocardium toward the first aortic arch pair (AA1). From 72 hpf onwards, the distal bound-ary was determined by the constriction of the OFT endocardium at the opening of the VA.

OFT width was assessed at 36 and 54 hpf on the 3D reconstructions obtained with Imaris (Bit-plane), and the diameter in the region of maximum BV constriction was determined (orange line in the figures). At 78 hpf, OFT expansion was measured in beating heart movies with the FiJi ImageJ software (*Schindelin et al., 2012*), in an average of 3–4 heartbeats per larva, with the following for-mula: OFT expansion (%) = [(maximum OFT area (during ventricular systole) - minimum OFT area (during ventricular diastole))/minimum OFT area] x 100.

## Dorsal aorta diameter measurement

Animals at 56 and 96 hpf were mounted on their side in 1% low-melting agarose as described above and imaged with an LSM700 Axio Imager with an LD LCI Plan-Apochromat 25×/0.8 lens. After the imaging at 56 hpf, the embryos were kept separately in 48-well plates until 96 hpf. After the second imaging at 96 hpf, the gDNA of individual fish was extracted, and the fish were genotyped after per-forming the measurements. The dorsal aorta diameter was measured after creating a maximum intensity projection of the images with the ZEN software. The diameter plotted is an average of four

measurements performed in different regions of the dorsal aorta, comprised within the 10 somites above the yolk extension.

## Data analysis and quantification

To count ECs in *Figure 3B* and SMCs in *Figure 4C*, *Tg(kdrl:EGFP)* and *TgBAC(pdgfrb:EGFP)*; *Tg(kdrl:TagBFP)* embryos and larvae, respectively, were fixed at 36, 72 or 75 hpf. The animals were then immunostained with anti-GFP and/or anti-tRFP antibodies and DAPI as described above. To count ECs, DAPI$^+$GFP$^+$ cells were counted in the OFT region highlighted in *Figure 3A*. To count SMCs, DAPI$^+$GFP$^+$ BFP$^-$ cells were counted in the OFT region. EC and SMC cell counting was performed with Imaris (Bitplane).

The quantification of the displacement length of photoconverted cells between 54 and 74 hpf was performed with Imaris (Bitplane) (*Figure 3L*). The distance of the photoconverted Kaede Red$^+$ cell from the bulbo-ventricular (BV) valve at 54 hpf was measured and then subtracted from the distance of the Kaede Red$^+$ cell farthest from the BV valve at 74 hpf.

For the quantification of Eln2$^+$ cells around the OFT, Eln2$^+$ DAPI$^+$ GFP$^-$ cells were counted. Quantification was started at the OFT mid-sagittal plane and cells were counted in two planes up and two down from that plane, at an increment of 2 μm per plane. An average of these five planes was used for the analyses. The percentage of Eln2$^+$ cells was calculated as (Eln2$^+$ DAPI$^+$ GFP$^-$ cells)/(DAPI$^+$ GFP$^-$ cells).

For quantification of p-Smad immunostaining, mean grey values of p-Smads and DAPI stainings were calculated with Fiji in at least seven cells from at least three single optical slices per embryo/larva. The cells analyzed were randomly chosen based on the DAPI signal, and ECs (or surrounding cells) were selected based on the *kdrl*:EGFP signal. Subsequently, p-Smad/DAPI ratios were generated for each cell, and the average of all the cells for each embryo was calculated. In *Figure 2B,C* and *Figure 2—figure supplement 1F*, the average of pSmad/DAPI normalized grey values was compared between different cells in the same embryo (ECs vs surrounding cells or OFT ECs vs heart ECs). The ratios obtained for each embryo were plotted in the graphs.

All statistical analyses were performed in GraphPad Prism (Version 6.07) and parametric *t*-tests or non-parametric Mann Whitney tests were performed after assessing the normal distribution of the data with D'Agostino-Pearson normality test. For multiple comparisons, one-way ANOVA, followed by Tukey's post-hoc test, or non-parametric Kruskal-Wallis test, followed by Dunn's post-hoc test, were performed. Illustrations were done in Inkscape (XQuartz X11).

## Randomization and blinding procedures

For all the experiments using *alk5* mutants, animals of different genotypes were collected, processed in the same dish (or tube) and genotyped after the imaging and analysis.

For all the experiments using Alk5 inhibitor, embryos deriving from *Tg(kdrl:EGFP)* outcrosses were collected. Afterwards, the embryos from the same batch were divided into two groups (treated and controls).

The investigators were blinded to allocation during experiments and outcome assessment, whenever possible.

## Transmission electron microscopy (TEM)

Larvae were collected at 72 hpf and the final portion of the trunk was cut and used for genotyping. The larvae were immediately fixed in ice-cold 1% PFA, 2% glutaraldehyde in 0.1 M sodium cacodylate buffer (pH 7.4) for 30 min on ice, and then stored at 4°C overnight. Samples were washed in 0.1 M sodium cacodylate buffer and postfixed in 2% (w/v) OsO$_4$, followed by *en bloc* staining with 2% uranyl acetate. Samples were dehydrated with a graded series of washes in acetone, transferred to acetone/Epon solutions, and eventually embedded in Epon. Ultra-thin sections (approximately 70 nm thick) obtained with a Reichert-Jung Ultracut E microtome were collected on copper slot grids. Sections were examined with a Jeol JEM-1400 Plus transmission electron microscope (Jeol, Japan), operated at an accelerating voltage of 120 kV. Digital images were recorded with an EM-14800 Ruby Digital CCD camera unit (3296px x 2472px).

## Heart isolation and transcriptome analysis

As the phenotype of *alk5* mutants is too mild to sort them at 56 hpf, we used the Alk5 inhibitor treatment (as described above) to obtain enough embryonic hearts for the transcriptomic analysis. Hearts from 56 hpf *Tg(kdrl:EGFP)* DMSO and Alk5 inhibitor treated embryos were manually dissected using forceps. Approximately 20 hearts per replicate were pooled, and total RNA was isolated using the miRNeasy micro kit, combined with on-column DNase digestion. Total RNA and library integrity were verified with LabChip Gx Touch 24 (Perkin Elmer). 100 ng of total RNA was used as input for VAHTS Stranded mRNA-seq Library (Vazyme) preparation, following manufacturer's protocol. Sequencing was performed on the NextSeq500 instrument (Illumina) using v2 chemistry, resulting in an average of 37M reads per library with $1 \times 75$ bp single end setup. The resulting raw reads were assessed for quality, adapter content and duplication rates with FastQC (available online at http://www.bioinformatics.babraham.ac.uk/ projects/fastqc). Reaper version 13–100 was used to trim reads with a quality drop below a mean of Q20 in a window of 10 nucleotides (*Davis et al., 2013*). Only reads between 30 and 150 nucleotides were used in subsequent analyses. Trimmed and filtered reads were aligned versus the Ensembl Zebrafish genome version DanRer11 (GRCz11.92) using STAR 2.4.0a with the parameter 'outFilterMismatchNoverLmax 0.1' to increase the maximum ratio of mismatches to mapped length to 10% (*Dobin et al., 2013*). The number of reads aligning to genes was counted with featureCounts 1.4.5-p1 tool from the Subread package (*Liao et al., 2014*). Only reads mapping at least partially inside exons were admitted and aggregated per gene, while reads overlapping multiple genes or aligning to multiple regions were excluded from further analyses. Differentially expressed genes were identified using DESeq2 version 1.18.1 (*Love et al., 2014*). To remove a batch effect from the comparison, the replicates were presented to DESeq2 as covariates (DMSO_1/Inhib_1 = 1, DMSO_2/Inhib_2 = 2). Genes were classified to be differentially expressed genes (DEG) with Benjamini-Hochberg corrected p-value<0.05 and $-0.59 \leq$ Log2 FC $\geq +0.59$. The Ensembl annotation was enriched with UniProt data (release 06.06.2014) based on Ensembl gene identifiers (Activities at the Universal Protein Resource (UniProt)).

The raw count matrix was batch corrected using CountClust (*Dey et al., 2017*) and then normalized with DESeq2. MA plots were produced to show DEG regulation per contrast.

Gene ontology enrichment of differentially expressed genes (log2FC>|1|; P adjusted <0.01) was calculated using cluster profiler (v 3.15.0) (*Yu et al., 2012*) and with annotations derived from org. Dr.eg.db (v 3.10.0). Main overrepresented biological processes were identified and agglomerated based on manual inspection of significant Gene Ontology database gene sets with p-value<0.01.

## RT-qPCR

Expression of *alk5a* and *alk5b* was analyzed in single 30 hpf larvae deriving from *alk5a* or *alk5b* heterozygous incrosses, respectively. DNA and RNA were extracted from single embryos using TRIzol reagent, followed by TRIzol-chloroform extracted, as previously described (*El-Brolosy et al., 2019*). Briefly, RNA was extracted from the aqueous phase and stored with isopropanol at −80°C. DNA was then subsequently precipitated from the organic phase after the addition of ethanol and, then, treated with proteinase K, to allow efficient genotyping by HRM. After the genotyping, RNA from 5 to 10 embryos of the same genotype was pooled and extracted by precipitation. 500 ng mRNA was used to synthesize cDNA using the Maxima First Strand cDNA kit.

To analyze *fbln5*, *efemp2b*, *sema3d*, *hmcn1* and *lum* expression, hearts were manually dissected from 72 hpf larvae. *alk5* mutant hearts were selected based on the OFT phenotype and 15 hearts were pooled per replicate. For *Figure 6—figure supplement 2C*, hearts were dissected and the body of the larva was genotyped. nine hearts were pooled according to the genotype. RNA was extracted as previously described for transcriptomic analyses.

For all the RT-qPCR experiments, DyNAmo ColorFlash SYBR Green qPCR Mix was used on a CFX connect Real-time System (Bio Rad).

All reactions were performed in technical triplicates and at least biological triplicates. For entire embryos qPCRs, every biological replicate consists of cDNA from a pool of 5–10 larvae per genotype, derived from independent crosses.

Gene expression values were normalized to the zebrafish *rpl13a* gene. Average Ct values for all conditions and details about statistical tests are listed in *Supplementary file 2*. Sequences of qPCR primers are listed in *Supplementary file 3*.

## Acknowledgements

We thank Radhan Ramadass for critical help with microscopy and all the fish facility staff for technical support, Rashmi Priya for the *myl7:mScarlet-Hsa.HRAS* plasmid and valuable suggestions, Matteo Perino for help with the analyses, discussions and critical comments on the manuscript, Sri Teja Mullapudi, Paolo Panza, Josephine Gollin, Felix Gunawan and Michelle Collins for suggestions and critical comments on the manuscript. Research in the Stainier lab is supported in part by the Max Planck Society, DFG (Sonderforschungsbereich) (SFB 834), the Leducq foundation, and the European Research Council (AdG project: ZMOD 694455).

## Additional information

### Competing interests

Didier YR Stainier: Senior Editor eLife. The other authors declare that no competing interests exist.

### Funding

| Funder | Grant reference number | Author |
|---|---|---|
| Max-Planck-Gesellschaft | | Giulia LM Boezio<br>Anabela Bensimon-Brito<br>Janett Piesker<br>Stefan Guenther<br>Christian SM Helker<br>Didier YR Stainier |
| Deutsche Forschungsgemeinschaft | SFB834 | Christian SM Helker<br>Didier YR Stainier |
| H2020 European Research Council | ZMOD 694455 | Didier YR Stainier |

The funders had no role in study design, data collection and interpretation, or the decision to submit the work for publication.

### Author contributions

Giulia LM Boezio, Conceptualization, Formal analysis, Validation, Investigation, Visualization, Methodology, Writing - original draft, Writing - review and editing; Anabela Bensimon-Brito, Conceptualization, Writing - review and editing; Janett Piesker, Investigation, Methodology, Writing - review and editing; Stefan Guenther, Formal analysis, Investigation, Methodology, Writing - review and editing; Christian SM Helker, Conceptualization, Supervision, Investigation, Project administration, Writing - review and editing; Didier YR Stainier, Conceptualization, Supervision, Funding acquisition, Writing - original draft, Project administration, Writing - review and editing

### Author ORCIDs

Giulia LM Boezio https://orcid.org/0000-0002-7776-7985
Anabela Bensimon-Brito http://orcid.org/0000-0003-1663-2232
Christian SM Helker http://orcid.org/0000-0003-0427-5338
Didier YR Stainier https://orcid.org/0000-0002-0382-0026

### Ethics

Animal experimentation: All zebrafish husbandry was performed under standard conditions, and all experiments were conducted in accordance with institutional (MPG) and national ethical and animal welfare guidelines (Proposal numbers: B2/1017, B2/1041, B2/1218, B2/1138). All procedures

conform to the guidelines from Directive 2010/63/EU of the European Parliament on the protection of animals used for scientific purposes.

## Decision letter and Author response
Decision letter https://doi.org/10.7554/eLife.57603.sa1
Author response https://doi.org/10.7554/eLife.57603.sa2

# Additional files

## Supplementary files
• Supplementary file 1. List of differentially expressed genes (>1 or <-1 log2FC) from RNA-seq dataset of 56 hpf dissected hearts from DMSO and Alk5 inhibitor-treated embryos.
• Supplementary file 2. Average Ct values of genes by RT-qPCR.
• Supplementary file 3. RT-qPCR primer list.
• Transparent reporting form

## Data availability
Sequencing data have been deposited in GEO under accession code GSE143770. All other data generated or analysed during this study are included in the manuscript, figures and source data files.

The following dataset was generated:

| Author(s) | Year | Dataset title | Dataset URL | Database and Identifier |
|---|---|---|---|---|
| Boezio GLM, Bensimon-Brito A, Piesker J, Guenther S, Helker CSM, Stainier DYR | 2020 | Identification of Alk5 targets zebrafish heart by transcriptomic analysis | https://www.ncbi.nlm.nih.gov/geo/query/acc.cgi?acc=GSE143770 | NCBI Gene Expression Omnibus, GSE143770 |

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
