## [Decision Letter]

**Acceptance summary:**

Endothelial cells have largely been neglected in vasculopathies, compared to smooth muscle cells and cardiac neural crests. This work now demonstrates a role of endothelial Alk5 signalling for the correct formation of the outflow tract in the zebrafish. It uncovers the extra-cellular matrix protein fbln5 as an effector, regulating the organization of the smooth muscle layer.

**Decision letter after peer review:**

Thank you for submitting your article "Endothelial TGF-β signaling instructs smooth muscle cell development in the cardiac outflow tract" for consideration by *eLife*. Your article has been reviewed by three peer reviewers, one of whom is a member of our Board of Reviewing Editors, and the evaluation has been overseen by Edward Morrisey as the Senior Editor. The following individuals involved in review of your submission have agreed to reveal their identity: Julien Vermot (Reviewer #2), Mohamad Azhar (Reviewer #3).

The reviewers have discussed the reviews with one another and the Reviewing Editor has drafted this decision to help you prepare a revised submission.

Summary:

This is an elegant and well written paper by Boezio et al. showing a role of the TGF-β receptor Alk5 in endothelial cells for the correct formation of the outflow tract/bulbus arteriosus in the zebrafish model. The authors assess the crosstalk between endothelial and smooth muscle cells. In newly generated double *alk5a/b* mutants, the authors show that the abnormal dilation of the outflow tract correlates with an increased number of endothelial cells followed by defective formation of the smooth muscle layer including the extra-cellular-matrix. The cardiovascular defects in *alk5* mutant fish being restricted to the outflow tract and ventral aorta, while all other vascular beds appear morphologically unaffected is remarkable. Transgenic rescue experiments demonstrate that *alk5* in endothelial cells is sufficient, but do not exclude an additional role in smooth muscle cells where *alk5* is also expressed. Transcriptomic analyses of pooled entire hearts, filtered with previous datasets, identify the extra-cellular matrix protein *fbln5* as a potential effector of *alk5*, which is supported by a non-targeted rescue experiment.

Based on advanced imaging and genetics, the fish appears as a relevant model to decipher potentially pathogenic mechanisms relevant to aorta aneurysm or dissection. Endothelial cells have largely been a neglected cell type in vasculopathies, with available literature predominantly focusing on the role of smooth muscle cells and cardiac neural crests.

However, several questions need to be addressed to support the main conclusions.

1) Alk5 expression. The main figure shows the expression of Alk5 at 78hpf. The rationale of this particular stage of development is not provided. The claim that "it becomes restricted to the OFT" suggests changes over time. This claim should be altered or more stages provided, the question of whether it is expressed at 36hpf when defects in replicating endothelial cells are shown should be raised. In addition, it is hard to visualize that Alk5 is restricted to the OFT based on the panels presented, it seems to have the same level of expression in the ventricle. A quantitative analysis would help to make the point. As the authors correctly mention in the Introduction, aortic smooth muscle cells, and not endothelial cells, are enriched for Alk5 expression in the mouse. Their data (Figure 1B-B") also show robust expression in smooth muscle cells. How do they interpret their results in endothelial cells in the light of these findings?

2) Alk5 signaling. The manuscript does not contain data showing any direct effect on TGF-β signaling. The authors have assumed all along that perturbing *alk5* levels would affect signaling but have not measured levels of phosphorylated Smad2/3 and Smad1/5 in endothelial cells to demonstrate this effect. While immunohistochemistry data for P-Smads would be ideal, the expression of some TGF-β target genes such as Pai1, etc., and reporter containing Smad binding elements (SBE) in isolated *alk5* mutant endothelial cells would be an alternative support of this conclusion. The authors have completely ignored the likely effects of *alk5* deletion on BMP signaling. Bmp2 has been shown to recruit Smad2 via Alk5/7, preferentially in the endothelial and embryonic cells, including in the dorsoventral axis formation in zebrafish (PMID: 24308972). Alternatively, Alk5 is also required for TGF-β's inhibitory effect on BMP signaling (22615489). Since Alk5 rests at the fulcrum of these two important signaling cascades with celebrated involvement in heart development, it would be essential to know in which direction the balance tilts in *alk5* mutants. This should be discussed in the manuscript and potentially supported by the transcriptomics data.

3) Alk5 mutants. What is the extent of sequence similarity between *alk5a* and *alk5b*? It is unclear why *alk5a* is so important to uncover a phenotype in the absence of *alk5b* if it is not highly expressed during embryogenesis and no compensation is observed between these genes in the respective mutants. Is there an explanation for this? Alk5 mutants display cardiac edema but it is unclear if the mutants display defective cardiac functions and at what stage? Since the data available in Video 3 is very limited, whether *alk5* mutants display normal beating of the heart and intracardiac blood flow remains unanswered.

4) In the transcriptomic dataset, how is Eln2, used in previous figures, and *alk1*, mentioned in the Discussion ? Why adhesion proteins, potentially relevant to the leaky endothelium or absence of the VA are not considered ?

5) Fbln5. The tissue layer where *fbln5* is expressed should be easily accessible using fluorescent ISH. Is *fbln5* expressed at 36hpf, before the smooth muscle coverage ? How is the VA in Alk5 mutants injected with *fbln5* mRNA ? The rescue of *eln2* organisation requires quantification.

6) The photoconversion experiments are amazing. Yet the conclusions may be overstated. It reads that *alk5* regulates cell displacement in the OFT. The term cell displacement may be a bit confusing as the VA is not forming in *alk5* mutant and it is unclear how this is actually related to the cell displacement observed in the control at this point. The VA absence could just arrest any morphogenetic processes. The mechanism of VA formation is not introduced and should be discussed.

Revisions expected in follow-up work:

1) Alk5 expression. To support the conclusion that "*alk5* expression becomes restricted to the OFT", several time points are required and especially from which time point the expression of Alk5 starts, and whether it is expressed at 36hpf when defects in replicating endothelial cells are shown.

2) Alk5 signaling. While immunohistochemistry data for P-Smads would be ideal, the expression of some TGF-β target genes such as Pai1, etc., and reporter containing Smad binding elements (SBE) in isolated *alk5* mutant endothelial cells would be an alternative support of this conclusion. The authors should consider likely effects of *alk5* deletion on BMP signaling.

3) Alk5 mutants. To what extent is *alk5* expression perturbed in the smooth muscle cells of the mutant fish? Also, is the smooth muscle cell proliferation defect rescued upon *alk5* overexpression in the mutants?

4) The transcriptomic dataset is validated for a single gene. The validation should be extended to the 5 genes selected in Figure 5C, as well as Alk5 responsive genes.

5) Fbln5. How is *fbln5* expression in the *alk5* mutants rescued with endothelial *alk5b*?

---

## [Author Response]

Revisions for this paper:1) Alk5 expression. The main figure shows the expression of Alk5 at 78hpf. The rationale of this particular stage of development is not provided. The claim that "it becomes restricted to the OFT" suggests changes over time. This claim should be altered or more stages provided, the question of whether it is expressed at 36hpf when defects in replicating endothelial cells are shown should be raised. In addition, it is hard to visualize that Alk5 is restricted to the OFT based on the panels presented, it seems to have the same level of expression in the ventricle. A quantitative analysis would help to make the point. As the authors correctly mention in the Introduction, aortic smooth muscle cells, and not endothelial cells, are enriched for Alk5 expression in the mouse. Their data (Figure 1B-B") also show robust expression in smooth muscle cells. How do they interpret their results in endothelial cells in the light of these findings?

We fully agree with the reviewers that information about the localization and timing of *alk5b* expression is important for the overall understanding of the mutant phenotype. We previously focused on its expression starting at 78 hpf because it is when the mutant phenotype becomes obvious, and also it is when the OFT becomes easily accessible for live imaging. Moreover, and unfortunately, the *alk5b* BAC reporter line displays an extremely low signal, consistent with the low expression of the gene, making the imaging – especially at early stages – particularly challenging. However, we have now been able to observe a weak signal throughout the heart at 36 hpf (green, Author response image 1).

**Author response image 1. respfig1:** *Tg(alk5b:*EGFP) expression (green) at 36 hpf.

In order to overcome these technical limitations, we used TGF-β signaling reporters. We imaged the Smad3 responsive element reporter line (*Tg(12xSBE:EGFP)*) – used as a proxy for Smad3 activation – and performed p-Smad3 antibody staining. With both of these approaches, we obtained evidence for TGF-β signaling in OFT ECs at 24 hpf (Figure 2 and Figure 2—figure supplement 1). Notably, p-Smad3 antibody staining showed an enrichment in OFT ECs at 24 hpf (Figure 2A-C), which, interestingly, was lost by 75 hpf (Figure 2B, Figure 2—figure supplement 1C, C’).

Based on these new findings, we propose that the early endothelial phenotype we observed in *alk5* mutants can be explained by the early Smad3 activation specifically in ECs. Subsequently, TGF-β signaling activity can be observed in SMCs. We speculate that a similar pattern might occur in mammals, where the early activation of TGF-β signaling in endothelial cells might have been overlooked. It is also worth noting that while *alk5*/Alk5 expression might be broad and weak, the expression of downstream factors might be more limited, thereby contributing to restrict the activation of the pathway.

2) Alk5 signaling. The manuscript does not contain data showing any direct effect on TGF-β signaling. The authors have assumed all along that perturbing alk5 levels would affect signaling but have not measured levels of phosphorylated Smad2/3 and Smad1/5 in endothelial cells to demonstrate this effect. While immunohistochemistry data for P-Smads would be ideal, the expression of some TGF-β target genes such as Pai1, etc., and reporter containing Smad binding elements (SBE) in isolated alk5 mutant endothelial cells would be an alternative support of this conclusion. The authors have completely ignored the likely effects of alk5 deletion on BMP signaling. Bmp2 has been shown to recruit Smad2 via Alk5/7, preferentially in the endothelial and embryonic cells, including in the dorsoventral axis formation in zebrafish (PMID: 24308972). Alternatively, Alk5 is also required for TGF-β's inhibitory effect on BMP signaling (22615489). Since Alk5 rests at the fulcrum of these two important signaling cascades with celebrated involvement in heart development, it would be essential to know in which direction the balance tilts in alk5 mutants. This should be discussed in the manuscript and potentially supported by the transcriptomics data.

We agree with the reviewers that assessing TGF-β signaling is a crucial point and thank them for the constructive suggestions. We have now performed p-Smad3 antibody staining at 24 hpf, i.e., before the onset of the phenotype, and detected a significant reduction of the signal in *alk5*-/- OFTs, in both ECs and surrounding cells (Figure 2D-F). We have also investigated the effect of Smad3 inhibition in OFT development (Figure 2G-K), and found that it is sufficient to induce a mild OFT phenotype in *alk5* heterozygotes (Figure 2G-H), and that it exacerbates the already severe phenotype in *alk5* mutants (Figure 2I-J).

We have also addressed in the Results section the potential role of Alk5 in regulating BMP signaling. Since the antibody staining for p-Smad1/5/8, now included in Figure 2—figure supplement 1F-I, did not show any significant enrichment in OFT ECs or significant difference between *alk5+/+* and *alk5*-/- OFTs, we propose that BMP signaling does not play a major role in OFT development, at least at early stages.

3) Alk5 mutants. What is the extent of sequence similarity between alk5a and alk5b? It is unclear why alk5a is so important to uncover a phenotype in the absence of alk5b if it is not highly expressed during embryogenesis and no compensation is observed between these genes in the respective mutants. Is there an explanation for this? Alk5 mutants display cardiac edema but it is unclear if the mutants display defective cardiac functions and at what stage? Since the data available in Video 3 is very limited, whether alk5 mutants display normal beating of the heart and intracardiac blood flow remains unanswered.

Alk5a and Alk5b display a very high level of sequence similarity (Author response image 2) in their intracellular domains (amino acids: 177-end), while they differ in their extracellular domains. *alk5a* mRNA levels are low at embryonic and larval stages; however, we have no information about protein levels in wild types or in *alk5b*-/- fish.

**Author response image 2. respfig2:** Clustal Omega (Madeira et al., 2019) protein alignment of Alk5a (B0EXP6_DANRE) and Alk5b (F1QQ20_DANRE).

Following the reviewers’ suggestion, we have now added information regarding *alk5-/-* cardiac function in the revised manuscript. We have measured the heart rate and found no significant differences even at 78 hpf, a stage when the OFT phenotype is already severe (Figure 1—figure supplement 1P), indicating that cardiac rhythm is not affected in *alk5* mutants. Nonetheless, at the same stage we observed in *alk5* mutants a consistent blood regurgitation in the AV canal compared with wild type (1/6 *alk5+/+*, 9/10 *alk5-/-*). We have now modified the Results section to include this information.

4) In the transcriptomic dataset, how is Eln2, used in previous figures, and alk1, mentioned in the Discussion ? Why adhesion proteins, potentially relevant to the leaky endothelium or absence of the VA are not considered ?

We agree with the reviewers that analyzing adhesion proteins will be interesting in future studies. However, in this work, we aimed to identify factors mediating the interaction between ECs and SMCs and, thus, selected proteins that could be secreted by ECs, or ligand-receptors which could mediate this cross-talk.

Following the reviewers’ comments, we have now measured in dissected 72 hpf *alk5+/+* and *alk5-/-* hearts the mRNA levels of the 5 genes selected from the intersection of the different datasets (Figure 6—figure supplement 1D). The new results can be found in Figure 6—figure supplement 1E. Moreover, we assessed *elnb* and *alk1* mRNA levels (Author response image 3) and found a slight downregulation in *alk5-/-* hearts, similar to what was observed in the RNAseq dataset on larval hearts from Alk5 inhibitor-treated animals (*elnb:* log2FC -2.41;; *alk1*: log2FC -0.48).

**Author response image 3. respfig3:** *elnb* and *alk1* expression in 72 hpf *alk5+/+* and *alk5-/-* hearts (p values from Mann Whitney tests).

In order to validate Alk5 responsive genes, we analyzed the Gene Ontology category “response to TGF-β” and only found genes belonging to the pathway (including *smad7, skia, skib, skilb, skila,* and *alk5b* itself), the expression of which has been shown to be reinforced in a positive feedback loop (Yan et al., 2018). Regarding TGF-β target genes commonly used in the literature, we observed a very diverse response in terms of expression in the RNA-seq dataset (Author response table 1), which might be due to secondary effects deriving from the OFT defects or to lack of tissue resolution (i.e., some genes might be TGF-β targets in some cardiac tissues and not in others). For all the above-mentioned genes, considering that we could not extract hearts and OFTs at 24-36 hpf (i.e., before the onset of the phenotype) due to technical limitations, we decided not to assess their mRNA levels at later stages, because of possible secondary effects.

However, for the sake of clarity, we added Supplementary file 1 with all the genes up- or down-regulated (>1 or <-1 log2FC) in the RNAseq.

**Author response table 1. resptable1:** Expression of some bona fide TGF-β targets in 56 hpf hearts from DMSO treated vs.Alk5 inhibitor treated larvae.

Gene	Base Mean DMSO	Base Mean Inhibitor	Log2FC Inhib/DMSO
*loxa*	750	353	-1.08
*ctgfa*	2672	1252	-1.09
*serpine1*	244	915	1.90
*acta2*	588	2171	1.88
*junba*	238	361	0.60
*junbb*	567	608	0.10

5) Fbln5. The tissue layer where *fbln5* is expressed should be easily accessible using fluorescent ISH. Is *fbln5* expressed at 36hpf, before the smooth muscle coverage ? How is the VA in Alk5 mutants injected with *fbln5* mRNA ? The rescue of eln2 organisation requires quantification.

We understand the reviewers’ questions about *fbln5* expression, and have now included new data showing that *fbln5* is indeed expressed in OFT ECs (Figure 6 and Figure 6—figure supplement 2).

1) We performed ISH and fluorescent ISH at 36 hpf, before SMC coverage of OFT, and observed *fbln5* expression in OFT ECs (Figure 6C-C’; Figure 6—figure supplement 2A-B’).

2) We generated a new EC-specific *Tg(fli1a:fbln5,EGFP)* line and found a partial rescue of a) OFT expansion, b) Eln2 organization, and c) VA formation in *alk5* mutants carrying this transgene (Figure 6D-K).

3) We performed additional RT-qPCR experiments and observed an increase in *fbln5* expression in *Tg(fli1a:alk5b-mScarlet) alk5-/-* hearts, compared with *alk5-/-* hearts not carrying the rescue transgene (Figure 6—figure supplement 2C).

Thus, these data strengthen our claim that Fbln5 can function downstream of Alk5 in ECs to regulate OFT development.

6) The photoconversion experiments are amazing. Yet the conclusions may be overstated. It reads that Alk5 regulates cell displacement in the OFT. The term cell displacement may be a bit confusing as the VA is not forming in *alk5* mutant and it is unclear how this is actually related to the cell displacement observed in the control at this point. The VA absence could just arrest any morphogenetic processes. The mechanism of VA formation is not introduced and should be discussed.

We thank the reviewers for the constructive comments and agree on the importance of distinguishing among these various possibilities. We appreciate that the time-lapse imaging alone is not a conclusive proof in terms of active cell movements, and that the lack of VA could be due to defects in other processes. Therefore, we have now refrained from using terms such as “migration” and “displacement”, and toned down our conclusions on the causes leading to the VA defects. We have rephrased the Results and Discussion sections accordingly.

Revisions expected in follow-up work:1) Alk5 expression. To support the conclusion that "alk5 expression becomes restricted to the OFT", several time points are required and especially from which time point the expression of Alk5 starts, and whether it is expressed at 36hpf when defects in replicating endothelial cells are shown.

Please see Revisions for this paper, point 1.

2) Alk5 signaling. While immunohistochemistry data for P-Smads would be ideal, the expression of some TGF-β target genes such as Pai1, etc., and reporter containing Smad binding elements (SBE) in isolated alk5 mutant endothelial cells would be an alternative support of this conclusion. The authors should consider likely effects of alk5 deletion on BMP signaling.

Please see Revisions for this paper, point 2.

3) Alk5 mutants. To what extent is alk5 expression perturbed in the smooth muscle cells of the mutant fish? Also, is the smooth muscle cell proliferation defect rescued upon alk5 overexpression in the mutants?

Please see Revisions for this paper, point 3.

4) The transcriptomic dataset is validated for a single gene. The validation should be extended to the 5 genes selected in Figure 5C, as well as Alk5 responsive genes.

Please see Revisions for this paper, point 4.

5) Fbln5. How is fbln5 expression in the alk5 mutants rescued with endothelial alk5b?

Please see Revisions for this paper, point 5.

References:

Madeira, F., Park, Y. M., Lee, J., Buso, N., Gur, T., Madhusoodanan, N., ... Lopez, R. (2019). The EMBL-EBI search and sequence analysis tools APIs in 2019. Nucleic Acids Res, 47(W1), W636-W641. doi:10.1093/nar/gkz268

Smyth, L. C. D., Rustenhoven, J., Scotter, E. L., Schweder, P., Faull, R. L. M., Park, T. I. H., and Dragunow, M. (2018). Markers for human brain pericytes and smooth muscle cells. J Chem Neuroanat*, 92*, 48-60. doi:10.1016/j.jchemneu.2018.06.001

Yan, X., Xiong, X., and Chen, Y. G. (2018). Feedback regulation of TGF-beta signaling. Acta Biochim Biophys Sin (Shanghai), 50(1), 37-50. doi:10.1093/abbs/gmx129